# The GTPase Nog1 co-ordinates the assembly, maturation and quality control of distant ribosomal functional centers

Purnima Klingauf-Nerurkar[1†], Ludovic C Gillet[2†], Daniela Portugal-Calisto[1], Michaela Oborská-Oplová[1,3], Martin Jäger[3], Olga T Schubert[2‡], Agnese Pisano[1], Cohue Peña[1], Sanjana Rao[1], Martin Altvater[3], Yiming Chang[3], Ruedi Aebersold[1,2§], Vikram G Panse[1*]

[1]Institute of Medical Microbiology, University of Zurich, Zurich, Switzerland; [2]Institute of Molecular Systems Biology, ETH Zurich, Zurich, Switzerland; [3]Institute of Biochemistry, ETH Zurich, Zurich, Switzerland

**\*For correspondence:**
vpanse@imm.uzh.ch

[†]These authors contributed equally to this work

**Present address:** [‡]Department of Human Genetics, University of California, Los Angeles, Los Angeles, United States; [§]Faculty of Science, University of Zurich, Zurich, Switzerland

**Competing interests:** The authors declare that no competing interests exist.

**Abstract** Eukaryotic ribosome precursors acquire translation competence in the cytoplasm through stepwise release of bound assembly factors, and proofreading of their functional centers. In case of the pre-60S, these steps include removal of placeholders Rlp24, Arx1 and Mrt4 that prevent premature loading of the ribosomal protein eL24, the protein-folding machinery at the polypeptide exit tunnel (PET), and the ribosomal stalk, respectively. Here, we reveal that sequential ATPase and GTPase activities license release factors Rei1 and Yvh1 to trigger Arx1 and Mrt4 removal. Drg1-ATPase activity removes Rlp24 from the GTPase Nog1 on the pre-60S; consequently, the C-terminal tail of Nog1 is extracted from the PET. These events enable Rei1 to probe PET integrity and catalyze Arx1 release. Concomitantly, Nog1 eviction from the pre-60S permits peptidyl transferase center maturation, and allows Yvh1 to mediate Mrt4 release for stalk assembly. Thus, Nog1 co-ordinates the assembly, maturation and quality control of distant functional centers during ribosome formation.

## Introduction

Error-free translation of the genetic code by the ribosome is critical for proteome homeostasis and cellular function. This essential task necessitates that only correctly assembled ribosomal subunits are committed for translation. Eukaryotic ribosome assembly is initiated by production of pre-rRNA in the nucleolus that is driven by RNA polymerase I (*Peña et al., 2017*; *Kressler et al., 2017*). The emerging pre-rRNA associates with small subunit (40S)-specific r-proteins, snoRNAs, U3snoRNP, numerous U3 proteins (UTPs) and assembly factors to form a pre-40S. Endonucleolytic cleavage releases the earliest pre-40S, permitting the remaining growing pre-rRNA to recruit large subunit (60S)-specific r-proteins and assembly factors to form a pre-60S. On their way through the nucleoplasm, pre-ribosomal particles interact with >200 assembly factors, including >40 energy-consuming AAA-ATPases, ABC-ATPases, GTPases and ATP-dependent RNA helicases. Elucidating their order of action and the mechanisms that co-ordinate their activities during pre-ribosome maturation is an important challenge.

Export-competent pre-ribosomes are transported through nuclear pore complexes (NPCs) into the cytoplasm, where they undergo maturation and proofreading before acquiring translation competence (*Nerurkar et al., 2015*). These steps include the release of assembly factors, transport receptors, pre-rRNA processing steps and the incorporation of remaining r-proteins that are critical for ribosome function. Cytoplasmic maturation of an exported pre-60S is initiated by the AAA-ATPase Drg1, which directly binds to and evicts the placeholder ribosomal-like protein Rlp24 from

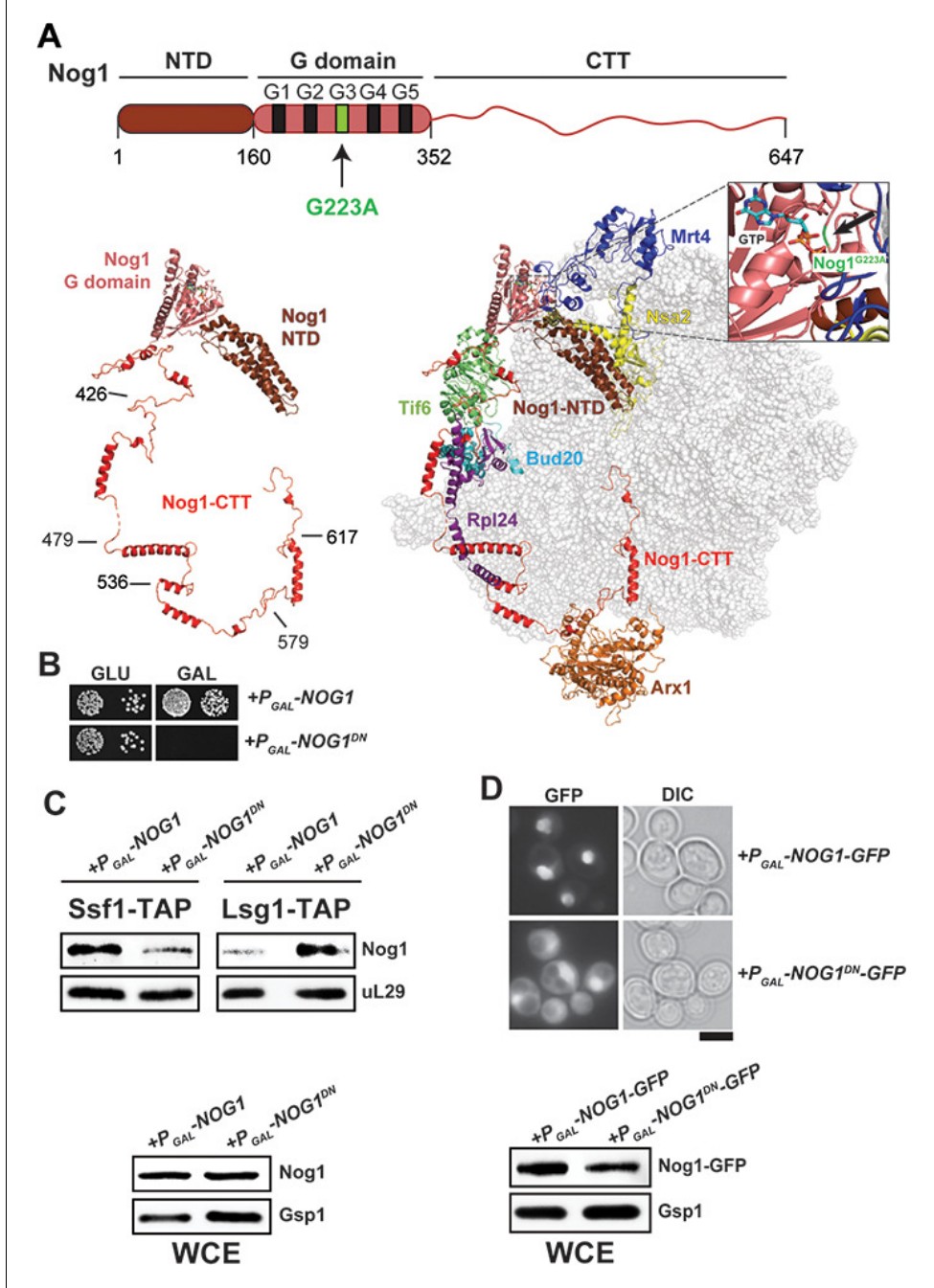

**Figure 1.** Nog1[DN] accumulates on a cytoplasmic pre-60S. (**A**) Domain organization and location of Nog1 on a pre-60S (PDB-3JCT). NTD: N-terminal domain; CTT: C-terminal tail. The black arrow points to the G223A mutation in the G3 motif within the G-domain of Nog1. Selected assembly factors (Mrt4, Nsa2, Tif6, Rlp24, Bud20 and Arx1) that are bound to the pre-60S are shown in color. (**B**) NOG1-G223A (NOG1[DN]) is dominant negative. Wild-type cells were transformed with a plasmid encoding NOG1 or NOG1[DN] under control of the galactose-inducible GAL1 promoter and spotted in 10-fold dilutions on glucose- or galactose-containing media. Plates were incubated at 30°C for 2–4 days. (**C**) Nog1[DN] accumulates on a cytoplasmic pre-60S. Early nucleolar (Ssf1-TAP) and cytoplasmic (Lsg1-TAP) particles were isolated from either Nog1- or Nog1[DN]-expressing cells. Co-enrichment of Nog1 was investigated by Western blotting. Whole-cell extracts (WCEs) depict Nog1 protein levels in the indicated strains. (**D**) Nog1[DN]-GFP accumulates in the cytoplasm. Wild-type cells expressing either Nog1-GFP or Nog1[DN]-GFP under control of GAL1 were grown in raffinose-containing synthetic medium to early log phase and then supplemented with 2% galactose. After 30 min incubation, cells were washed, incubated in YPD for 3 hr and then

*Figure 1 continued on next page*

*Figure 1 continued*

visualized by fluorescence microscopy. Scale bar = 5 μm. Quantification is listed in **Supplementary file 3**. WCEs depict Nog1 protein levels in the indicated strains.

the pre-60S (*Pertschy et al., 2007*; *Kappel et al., 2012*; *Kressler et al., 2012*). This step is a prerequisite for the release of two GTPases (Nog1 and Nug1), the assembly factor Nsa2, the subunit anti-association factor Tif6, the ribosomal-like protein Mrt4 and the export factors Mex67-Mtr2, Bud20, Arx1 and Nmd3 (*Loibl et al., 2014*; *Zisser et al., 2018*). Following Rlp24 release, the 60S maturation pathway, via a yet unknown mechanism, bifurcates to proofread the polypeptide exit tunnel (PET) and to assemble the ribosomal stalk (*Lo et al., 2010*). PET maturation is accomplished through the release of Arx1, an assembly factor that covers the tunnel and prevents the premature loading of the protein-folding chaperone machinery (*Bradatsch et al., 2012*; *Greber et al., 2012*).

Arx1 release from the pre-60S requires the cytoplasmic zinc-finger protein Rei1, the DnaJ domain-containing protein Jjj1 and Ssa1/Ssa2 (Hsp70) ATPase activity (*Meyer et al., 2007*; *Meyer et al., 2010*). Cryo-electron microscopy (cryo-EM) studies have revealed that Rei1 inserts its C-terminal tail (Rei1-CTT) into the PET (*Greber et al., 2016*). Failure to insert the Rei1-CTT into the tunnel impairs Arx1 release, and blocks subsequent pre-60S maturation (*Greber et al., 2016*). Cryo-EM analyses of a nuclear pre-60S revealed that the C-terminal tail of the GTPase Nog1 (Nog1-CTT) intertwines around Rlp24, makes contact with Arx1 and inserts its end into the PET (*Wu et al., 2016*). For Rei1-CTT to gain access to the PET, the Nog1-CTT end must be extracted. How these events are orchestrated during cytoplasmic maturation is unknown.

Another functional center whose assembly is completed in the cytoplasm is the 60S ribosomal stalk. The stalk is built from a single copy of ribosomal proteins uL10 (Rpp0) and two heterodimers of P1 (Rpp1) and P2 (Rpp2), and plays an essential role by recruiting and activating translation elongation factors. On a mature 60S subunit, the stalk is anchored through the interaction of uL10 with rRNA and uL11 (Rpl12) (*Ben-Shem et al., 2011*). During nuclear 60S assembly, the ribosomal-like protein Mrt4 functions as a placeholder for uL10 (*Lo et al., 2009*; *Kemmler et al., 2009*). Mrt4 removal from the pre-60S is triggered by the release factor Yvh1 (*Kemmler et al., 2009*; *Lo et al., 2009*). uL10 can be loaded onto the pre-60S to assemble the stalk only after Mrt4 is released. How stalk assembly is coordinated with other cytoplasmic maturation steps remains unclear.

Removal of the placeholders Rlp24, Arx1 and Mrt4 from the pre-60S is an essential prerequisite for PET maturation and quality control, polypeptidyl transferase center (PTC) maturation, stalk assembly and the incorporation of the remaining r-proteins. How these spatially distant events on the pre-60S are orchestrated remains unknown. By employing quantitative mass spectrometry, genetic and cell-biological approaches, we reveal that the ATPase Drg1 and the GTPase Nog1 license these events, and ensure that only correctly assembled 60S subunits enter translation.

## Results

The nuclear localized GTPase Nog1 co-enriches with a late pre-60S that contains the export factors Nmd3, Bud20, Mex67-Mtr2 and Arx1 (*Jensen et al., 2003*; *Kallstrom et al., 2003*). In wild-type (WT) cells, Nog1 is not detected on a cytoplasmic pre-60S purified through tandem-affinity purification (TAP) of Lsg1 (*Kressler et al., 2008*; *Altvater et al., 2012*). However, Nog1 mislocalizes to the cytoplasm, and accumulates on Lsg1-TAP particles upon impairment of the cytoplasmic AAA-ATPase Drg1 or upon treating yeast cells with the Drg1-inhibitor diazaborine (DIA) (*Pertschy et al., 2007*; *Loibl et al., 2014*). These data show that Nog1 travels to the cytoplasm, where it is released in a Drg1-dependent manner and rapidly recycled back to the nucleus. How and exactly when Nog1 is released from the pre-60S in the cytoplasm is unclear.

## Nog1$^{DN}$ accumulates on a cytoplasmic pre-60S particle

The Nog1 G-domain exhibits characteristic G1–G5 motifs (*Figure 1A*, upper panel) suggesting that GTP-binding or hydrolysis of Nog1, like that of other GTPases such as Nug2/Nog2 and Lsg1, might regulate interactions between Nog1 and the pre-60S particle (*Hedges et al., 2005*; *Matsuo et al., 2014*). Dominant-negative mutations have been described within the G1 motif of Lsg1(K349N/R/T) (*Hedges et al., 2005*) and the G3 motif of Nug2 (G369A) (*Bourne et al., 1991*; *Hedges et al.,*

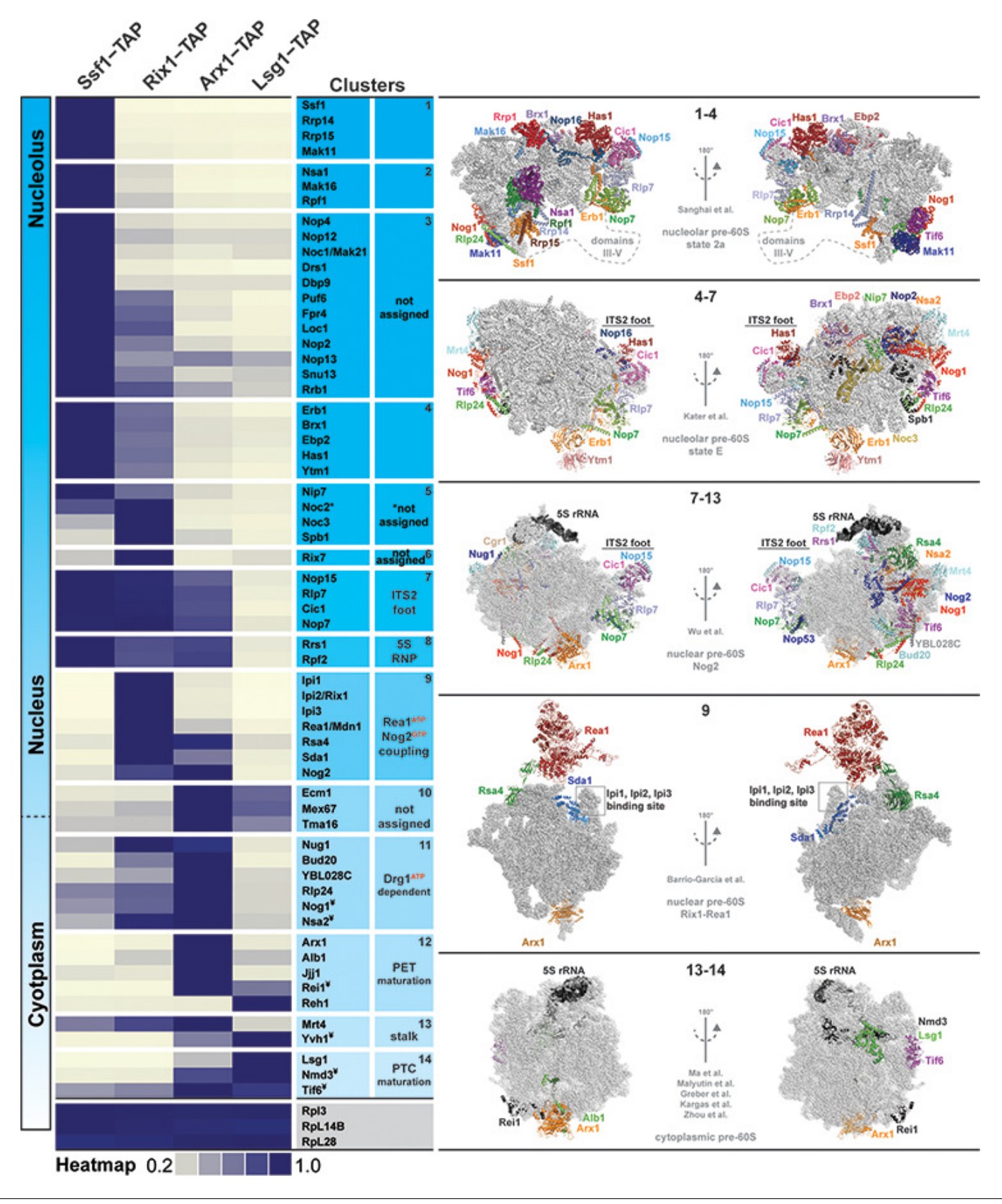

**Figure 2.** Association of assembly factors with pre-60S at different maturation stages revealed by SWATH-MS. Pre-60S particles representing nucleolar (Ssf1-TAP), nuclear (Rix1-TAP), nuclear to cytoplasmic (Arx1-TAP) and cytoplasmic (Lsg1-TAP) stages were analyzed by SWATH-MS. A heat map was generated on the basis of the average of three independent biological replicates and depicts the relative and individual enrichment of assembly factors with maturing pre-60S particles. The acquired SWATH data were normalized on the basis of the average intensities of three depicted 60S r-proteins, and the intensity of each factor was scaled to the highest intensity of that factor in the selected condition represented in the heat map. Maximum enrichment is depicted in purple and minimum enrichment in gold. Assembly factors with similar maximum enrichment profiles were then manually grouped into clusters on the basis of the available cryo-EM data (nucleolar pre-60S: PDB-6C0F and PDB-6ELZ; nuclear pre-60S: PDB-3JCT and PDB-5FL8; cytoplasmic pre-60S: PDB-5H4P, PDB-5T62, PDB-5APN, PDB-6RZZ and PDB-6N8N). ¥ depicts factors whose protein levels were analyzed by Western blotting (*Figure 2—figure supplement 1*).

The online version of this article includes the following figure supplement(s) for figure 2:

**Figure supplement 1.** Co-enrichment of assembly factors with a maturing pre-60S subunit.

---

2005; *Matsuo et al., 2014*) that impair the release of these GTPases from a pre-60S. A G224A mutation in the G3 motif of human Nog1, which presumably blocks GTP hydrolysis, was shown to be dominant negative in mammalian cells, and induced pre-rRNA processing and assembly defects (*Lapik et al., 2007*).

We investigated whether the G-domain contributes to Nog1 release from the pre-60S in the cytoplasm. To this end, the orthologous mutation in the G3 motif (G223A) of yeast Nog1 (*Figure 1A*), hereafter termed Nog1[DN], was transformed into a Nog1 shuffle strain, wherein the *NOG1* gene was disrupted but the viability of the yeast cells was maintained through a centromeric plasmid containing a WT copy of *NOG1*. We did not obtain transformants for the Nog1[DN] mutant. To demonstrate dominant-negative behavior, we placed the Nog1[DN] mutant under the control of an inducible *GAL1* promoter, and transformed this plasmid into WT yeast cells. On glucose-containing medium, where Nog1[DN] expression is repressed, the resulting transformants grew similar to WT. By contrast, expression of Nog1[DN] in galactose-containing medium was lethal to yeast cells (*Figure 1B*), confirming the dominant-negative behavior of the G223A mutation.

Nog1 is recruited to the pre-60S in the nucleolus (*Kressler et al., 2008*; *Altvater et al., 2012*), and is released from the particle in the cytoplasm (*Pertschy et al., 2007*; *Lo et al., 2010*; *Altvater et al., 2012*). We investigated whether the Nog1[DN] mutant was released from the pre-60S in the cytoplasm. For this, we isolated the Lsg1-TAP particle after inducing expression of either Nog1 or the Nog1[DN] mutant allele for 2.5 hr (*Figure 1C*). Western analyses revealed that Nog1[DN] mutant protein, but not Nog1, accumulated on the Lsg1-TAP particle (*Figure 1C*). Whole cell extracts (WCE) revealed similar Nog1 and Nog1[DN] protein levels (*Figure 1C*), suggesting that Nog1[DN] co-enrichment with Lsg1-TAP is not due to altered expression of the mutant protein. Moreover, the Nog1[DN]-GFP fusion showed an increase in cytoplasmic signal, supporting the notion that Nog1[DN] release from the pre-60S in the cytoplasm is impaired (*Figure 1D*). Although a nuclear signal of Nog1[DN]-GFP is observed in these cells, this mutant did not efficiently co-enrich with Ssf1-TAP under the same conditions (*Figure 1C*), possibly owing to blockage of downstream cytoplasmic maturation steps that indirectly impair early assembly steps (see later). We conclude that a functional G-domain is essential to evict Nog1 from the pre-60S in the cytoplasm.

## Nog1[DN] impairs cytoplasmic maturation of the pre-60S particle

We investigated the consequences of impaired Nog1[DN] release on the composition of the cytoplasmic Lsg1-TAP particle by Sequential Window Acquisition of all THeoretical fragment ion spectra mass spectrometry, also termed SWATH-MS. SWATH-MS is a mass spectrometry approach that combines data-independent acquisition with a peptide-centric data query strategy (*Gillet et al., 2012*). In contrast to selected reaction monitoring mass spectrometry (SRM-MS) (*Picotti and Aebersold, 2012*), SWATH-MS can be extended to the analysis of any peptide and protein of interest post-acquisition, while maintaining optimal consistency of quantification in pull-down samples (*Collins et al., 2013*; *Lambert et al., 2013*).

We interrogated quantitatively the protein composition of four well-characterized pre-60S particles representing different maturation stages (*Nissan et al., 2002*): Ssf1-TAP, an early nucleolar particle; Rix1-TAP, a nucleoplasmic particle; Arx1-TAP, a particle loaded with nuclear export factors; and Lsg1-TAP, an exclusively cytoplasmic pre-60S. The data were analyzed using OpenSWATH software (*Röst et al., 2014*), and accuracy was compared with that of SRM-MS based analyses

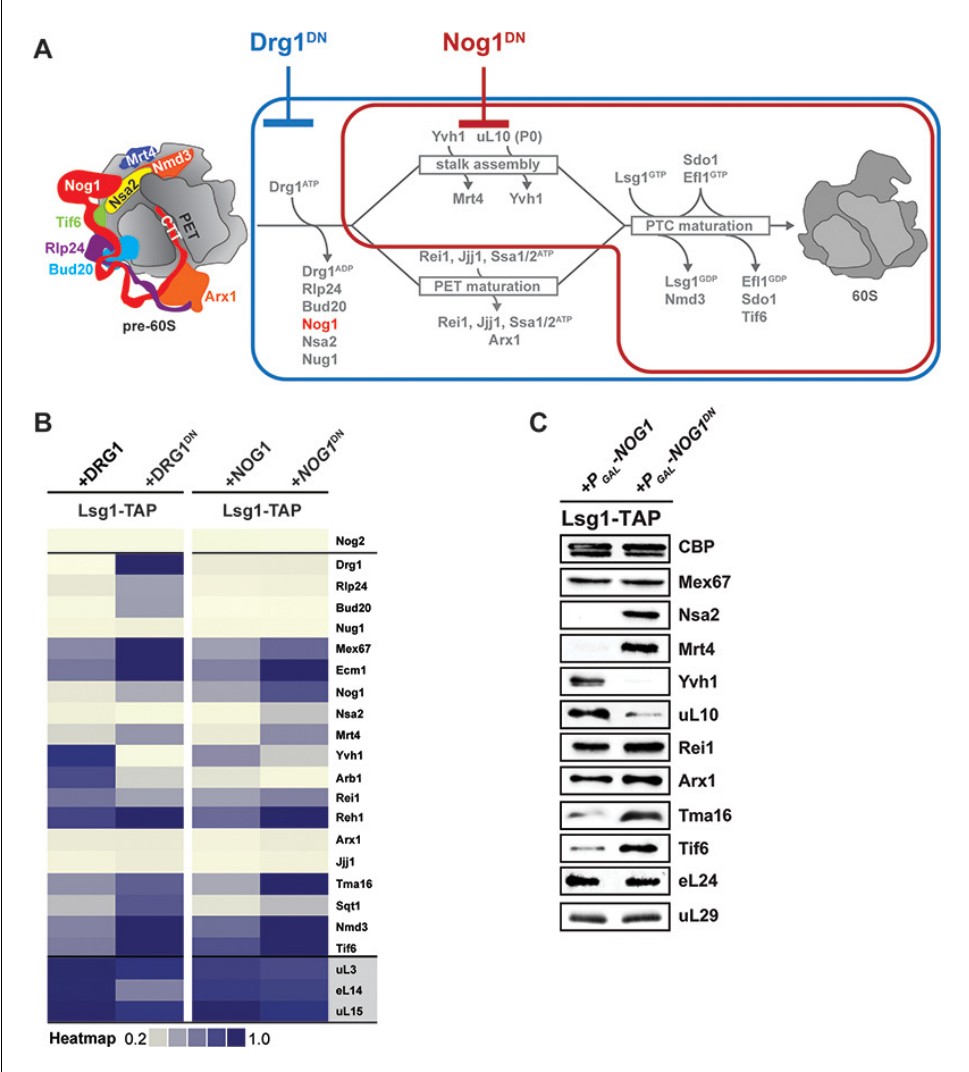

**Figure 3.** Nog1[DN] impairs cytoplasmic maturation of a pre-60S. (**A**) Current model for the cytoplasmic maturation pathway of a pre-60S. A cytoplasmic pre-60S at different maturation stages is represented as a cartoon. Energy-consuming assembly factors are indicated with ATP or GTP. Events that are impaired by Drg1[DN] and Nog1[DN] expression are highlighted in blue and red, respectively. (**B**) SWATH-MS analysis of Lsg1-TAP pre-60S in Drg1[DN]- and Nog1[DN]-expressing cells. Assembly factors accumulating on the pre-60S particle in a Drg1- or Nog1-dependent manner were clustered on the basis of their increased enrichment. (**C**) Western analyses of selected pre-60S assembly factors involved in cytoplasmic maturation. Nog1[DN]-trapped Lsg1-TAP particles were subjected to Western analysis using the indicated antibodies. CBP: calmodulin-binding peptide present in the Lsg1 TAP-tag.

(*Altvater et al., 2012*). We found that the proteomic heat map obtained using SWATH-MS was in agreement with that generated through SRM-MS (*Altvater et al., 2012*) and Western analyses (*Figure 2—figure supplement 1*). In contrast to SRM-MS, SWATH-MS permitted the quantitation of the approximate residence time of nearly all assembly factors along the 60S maturation pathway (*Figure 2*).

We correlated the protein heat map with reported pre-ribosome cryo-EM structures (*Sanghai et al., 2018*; *Kater et al., 2017*; *Wu et al., 2016*; *Barrio-Garcia et al., 2016*; *Ma et al., 2017*; *Malyutin et al., 2017*; *Greber et al., 2016*). These analyses allowed organization of assembly factors into different clusters on a maturing pre-60S. For example, the early nucleolar Ssf1-Rrp14-Rrp15-Mak11 cluster, the nuclear ITS2 factors Nop15-Rlp7-Cic1-Nop7, the 5S RNP-associated Rrs1-Rpf2, the Rix1-Rea1 machinery and the Drg1-dependent factors Rlp24-Nog1-Nsa2-Bud20 appear to undergo coordinated and grouped release. Our analyses also revealed temporal association of assembly factors for which structural information in the context of the pre-60S is lacking. These

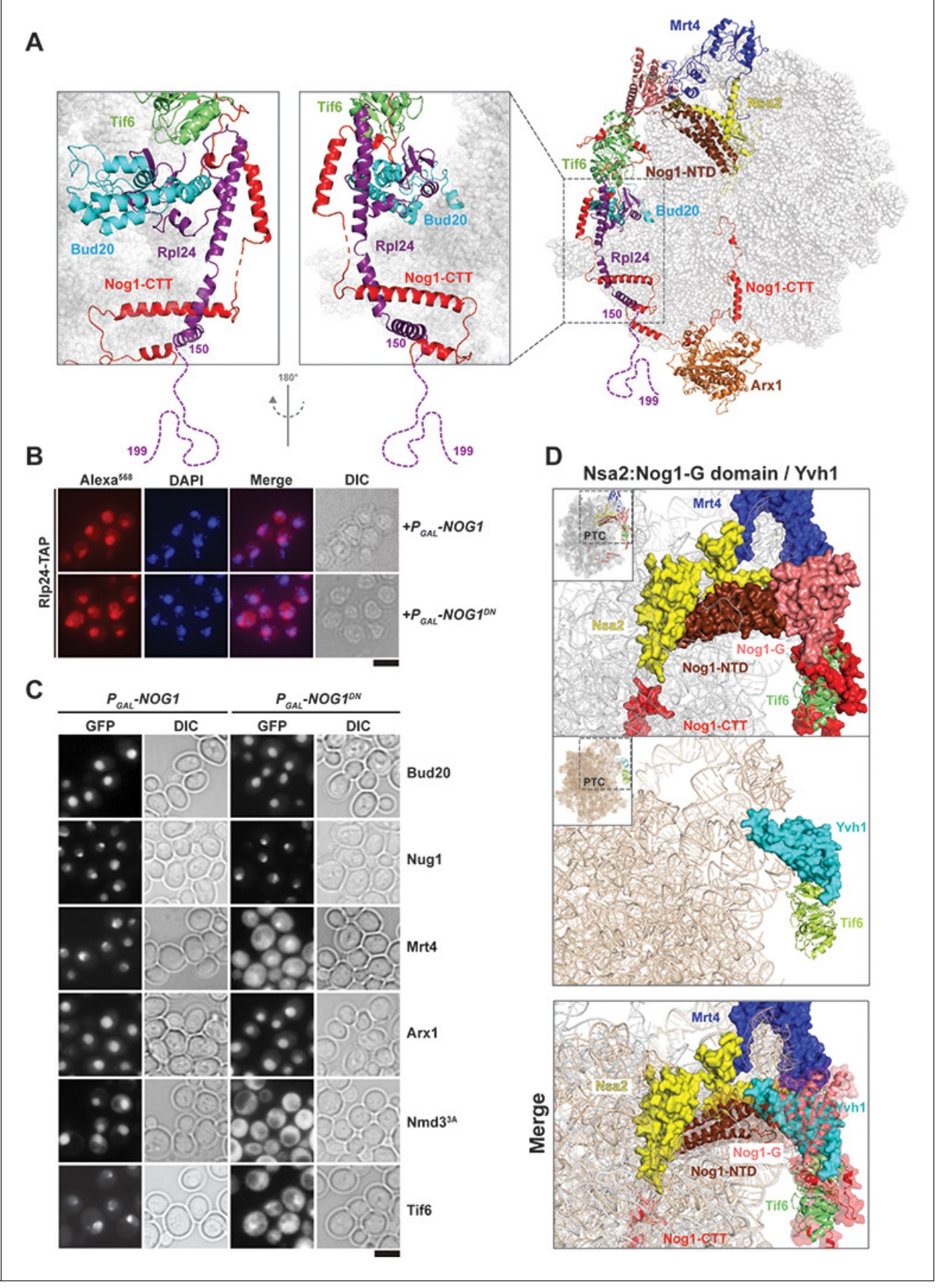

**Figure 4.** Nog1$^{DN}$ impairs stalk assembly and terminal maturation steps. (**A**) Cryo-EM structure (PDB-3JCT) of a Nog1-containing pre-60S particle, depicting the interaction between Rlp24 and Nog1. (**B**) A RLP24-TAP strain expressing NOG1 or NOG1$^{DN}$ under the control of GAL1 was grown in raffinose-containing synthetic medium to early log phase, then supplemented with 2% galactose, and further grown for 3 hr. Cells were then fixed and prepared for indirect immunofluorescence using an anti-Protein A antibody. Cells were visualized by fluorescence microscopy. Scale bar = 5 μm. Quantification is listed in **Supplementary file 3**. (**C**) The indicated GFP-tagged strains expressing NOG1 or NOG1$^{DN}$ under control of GAL1 were grown as in panel (B) and then visualized by fluorescence microscopy. Scale bar = 5 μm. Quantification of the data is listed in Supplementary Table 2. (**D**) Yvh1 clashes with the Nsa2:Nog1-G domain. Cryo-EM structures of nuclear Nog1-containing (PDB-3JCT) and late cytoplasmic Yvh1-containing (PDB-6RZZ) pre-60S particles were superimposed. Release of the Nsa2:Nog1-G domain complex allows recruitment of Yvh1 to the pre-60S.

factors include nucleolar proteins and the uncharacterized nuclear-localized Tma16 (*Figure 2*). We conclude that SWATH-MS is a reliable tool to quantify the protein contents of pre-ribosomal particles.

Next, we quantified and compared the protein contents of genetically trapped cytoplasmic pre-60S particles isolated from yeast cells expressing either the dominant-negative Drg1-E617Q (Lsg1-TAP:Drg1$^{DN}$) (*Altvater et al., 2012*) or the Nog1$^{DN}$ (Lsg1-TAP:Nog1$^{DN}$) mutant. We focused on those assembly factors that are known to participate in pre-60S cytoplasmic maturation (*Figure 3A*). In agreement with previous studies (*Pertschy et al., 2007*; *Lo et al., 2010*; *Altvater et al., 2012*), a Drg1$^{DN}$:Lsg1-TAP particle accumulated Rlp24, Bud20, Nog1, Mrt4, Nmd3 and Tif6 (*Figure 3A* and *Figure 3B*). In addition, the cytoplasmic chaperone Sqt1, which loads uL16 onto the pre-60S (*Pausch et al., 2015*), and the uncharacterized ribosome-associated factor Tma16 (*Fleischer et al., 2006*) accumulated on the Drg1$^{DN}$:Lsg1-TAP particle (*Figure 3B*). Like the Drg1$^{DN}$:Lsg1-TAP particle, the Lsg1-TAP:Nog1$^{DN}$ particle accumulated Mrt4, Sqt1, Tma16, Nmd3 and Tif6 (*Figure 3A* and *Figure 3B*). However, unlike the Drg1$^{DN}$:Lsg1-TAP particle, Rlp24 and its interaction partner Bud20 did not accumulate on the Lsg1-TAP:Nog1$^{DN}$ particle (*Figure 3A* and *Figure 3B*). Both Drg1$^{DN}$ and Nog1$^{DN}$-trapped Lsg1-TAP particles failed to recruit Yvh1. Although Rei1 recruitment to the Lsg1-TAP:Drg1$^{DN}$ particle was impaired, Nog1$^{DN}$-trapped particles recruited Rei1 (*Figure 3C*). By contrast, Reh1, which functionally overlaps with Rei1 (*Parnell and Bass, 2009*), accumulated on both Drg1$^{DN}$ and Nog1$^{DN}$-trapped Lsg1-TAP particles (*Figure 3B*). The relative co-enrichments of assembly factors on the Lsg1-TAP:Nog1$^{DN}$ particle monitored by SWATH-MS were in agreement with Western analyses (*Figure 3C*). These data suggest that Nog1$^{DN}$-expressing cells are impaired in a subset of events along the 60S cytoplasmic maturation pathway.

## Nog1$^{DN}$ does not hinder initiation of cytoplasmic maturation

Cryo-EM studies have revealed that different domains of the assembly factor Nog1 (NTD, G-domain and CTT) meander from the peptidyl transferase center (PTC) to the PET, and contact assembly factors on the pre-60S (*Wu et al., 2016*). The Nog1-NTD interacts with the assembly factor Nsa2, whereas the Nog1-G domain contacts the ribosomal-like protein Mrt4, and the Nog1-CTT intertwines around Rlp24 and contacts Arx1 (*Figure 1A*). Although Nog1, Nsa2, Arx1, Bud20, Mrt4 and Rlp24 are already recruited to the pre-60S during nucleolar/nucleoplasmic maturation, this cluster of assembly factors (*Figure 1A*) is evicted only in the cytoplasm (*Figure 3A*) (*Pertschy et al., 2007*; *Lo et al., 2010*; *Altvater et al., 2012*).

To dissect the impact of impaired Nog1$^{DN}$ release from the pre-60S on the cytoplasmic maturation pathway, we employed a cell-biological approach. Release of the ribosomal-like-protein Rlp24 from the pre-60S initiates the cytoplasmic maturation cascade (*Figure 3A*) (*Pertschy et al., 2007*; *Lo et al., 2010*). On the pre-60S, Rlp24 directly contacts Bud20, and then intertwines around the Nog1-CTT, forming a Nog1-CTT:Rlp24 complex (*Figure 4A*). The last 40 residues of the Rlp24 C-terminal tail (155–190), which remain unresolved in cryo-EM structures (*Wu et al., 2016*; *Zhou et al., 2019*), recruit and activate the AAA-ATPase Drg1 to extract Rlp24 from the pre-60S (*Pertschy et al., 2007*; *Lo et al., 2010*; *Kappel et al., 2012*; *Altvater et al., 2012*). Given the intimate interactions between Nog1-CTT and Rlp24 on the pre-60S (*Wu et al., 2016*) (*Figure 4A*), we wondered whether Nog1$^{DN}$ hinders Drg1-mediated Rlp24 release. As judged by immunofluorescence, Rlp24-TAP did not mislocalize to the cytoplasm upon Nog1$^{DN}$ expression (*Figure 4B*), suggesting that Nog1$^{DN}$ accumulation on a cytoplasmic pre-60S did not interfere with the release and recycling of Rlp24 back to the nucleus. These data are consistent with those from SWATH-MS analyses, which show that Rlp24 does not accumulate on the Lsg1-TAP:

Nog1$^{DN}$ particle (*Figure 3B*). This finding is in agreement with previous studies which showed that Drg1-ATPase activity is necessary and sufficient to release Rlp24 from the pre-60S (*Kappel et al., 2012*). The presence of Nog1$^{DN}$ on the pre-60S did not interfere with recruitment of r-protein eL24 (yeast Rpl24), as judged by Western analyses (*Figure 3C*). Cell-biological studies indicate that the release of the Rlp24-interacting assembly factor Bud20 from the pre-60S is also not impaired in Nog1$^{DN}$-expressing cells (*Figure 4C*). We conclude that Nog1$^{DN}$ does not disturb initiation of the pre-60S cytoplasmic maturation cascade.

While the Lsg1-TAP:Drg1$^{DN}$ particle accumulated Rlp24, Drg1$^{DN}$, Nog1 and another GTPase Nug1 (*Altvater et al., 2012*), the Lsg1-TAP:Nog1$^{DN}$ particle accumulated only Nog1$^{DN}$, and not Rlp24, Drg1 or Nug1 (*Figure 3B* and *Figure 4C*). On the basis of these data, we conclude that Nog1$^{DN}$ eviction from the pre-60S occurs after Drg1-mediated Rlp24 release.

## Nog1$^{DN}$ interferes with ribosomal stalk assembly

Following Drg1-mediated Rlp24 release from the pre-60S, the cytoplasmic maturation pathway bifurcates to: (1) assemble the ribosomal stalk, and (2) mature the PET and proofread its integrity (*Figure 3A*) (*Lo et al., 2010*). The ribosomal stalk is assembled in the cytoplasm from the acidic r-proteins uL10 (yeast P0) and P1/P2 heterodimers. Mrt4 functions as a nuclear placeholder for uL10 and joins the pre-60S in the nucleolus (*Lo et al., 2010*; *Kemmler et al., 2009*). Mrt4 release in the cytoplasm requires Yvh1, an assembly factor that maximally co-enriches with Lsg1-TAP (*Kemmler et al., 2009*; *Lo et al., 2010*). SWATH-MS and Western analyses revealed that Lsg1-TAP: Nog1$^{DN}$ failed to recruit Yvh1, and accumulated Mrt4 (*Figure 3B* and *Figure 3C*). In agreement with these data, we found that Nog1$^{DN}$-expressing cells mislocalized Mrt4-GFP into the cytoplasm (*Figure 4C*). Accordingly, Western analyses showed that uL10 recruitment to Lsg1-TAP:Nog1$^{DN}$ was impaired (*Figure 3C*). Cryo-EM studies suggest that the Yvh1-binding site on the pre-60S clashes with the Nog1-G domain (*Figure 4D*) (*Zhou et al., 2019*; *Kargas et al., 2019*), providing an explanation as to why Nog1 eviction is critical for stalk assembly. Given that Drg1-ATPase activity is required to evict Nog1, these structural data explain why Drg1$^{DN}$-expressing cells are impaired in Mrt4 release, and consequently stalk assembly (*Altvater et al., 2012*; *Lo et al., 2010*). We conclude that stalk assembly requires Drg1-ATPase activity and a functional Nog1-GTPase domain.

## Nog1$^{DN}$ does not impair PET maturation

We investigated the impact of Nog1$^{DN}$ expression on PET maturation and quality control (*Figure 3A*). During nuclear assembly of the pre-60S, the aminopeptidase fold of Arx1 covers the PET (*Bradatsch et al., 2012*; *Greber et al., 2012*). Arx1 eviction from the pre-60S is triggered in the cytoplasm by Rei1 (*Meyer et al., 2010*). Rei1 is recruited to the pre-60S through its N-terminal domain (*Lebreton et al., 2006*), whereas its C-terminal tail-end (Rei-CTT) is inserted into the PET (*Greber et al., 2016*). Rei1 recruitment to the pre-60S requires Drg1-mediated Rlp24 release (*Figure 5A*) (*Lo et al., 2010*; *Altvater et al., 2012*). We found that Nog1$^{DN}$ expression did not interfere with Rei1 recruitment (*Figure 3B* and *Figure 3C*). However, co-expression of Nog1$^{DN}$ and Drg1$^{DN}$ impaired Rei1 recruitment to the pre-60S (*Figure 5A*). Accordingly, Nog1$^{DN}$-expressing cells when treated with the Drg1-inhibitor diazaborine (DIA) mislocalized Arx1-GFP to the cytoplasm in these cells (*Figure 5B*). Cryo-EM studies indicate that the Rei1-binding site on the pre-60S clashes with the Nog1-CTT:Rlp24 complex (*Figure 5C*) (*Zhou et al., 2019*), providing an explanation as to why PET maturation is inhibited in Drg1$^{DN}$-expressing (or DIA-treated) cells. All of these data show that Rlp24 extraction from the Nog1-CTT is critical to recruit Rei1 to the pre-60S in order to initiate PET maturation.

Nog1$^{DN}$-expressing cells did not mislocalize Arx1-GFP into the cytoplasm (*Figure 5D*), suggesting that Nog1$^{DN}$ presence on the pre-60S does not interfere with the ability of Rei1-CTT to probe the PET. Failure to insert Rei1-CTT into the PET by attaching a bulky domain (*rei1-TAP*) blocks Arx1 release and progression of cytoplasmic maturation, suggesting that PET proofreading represents a quality control check point (*Greber et al., 2016*). Consistent with this, Nog1$^{DN}$ expression in a *rei1-TAP* mutant (in which Rei1-CTT function in probing the PET is compromised) mislocalized Arx1-GFP to the cytoplasm (*Figure 5D*). Before Rei1-CTT can probe PET integrity, the terminal end of Nog1-CTT needs to be extracted from the PET. Given that Rei1 recruitment and PET probing is not impaired in Nog1$^{DN}$-expressing cells, we suggest that Nog1$^{DN}$-CTT has been removed from the

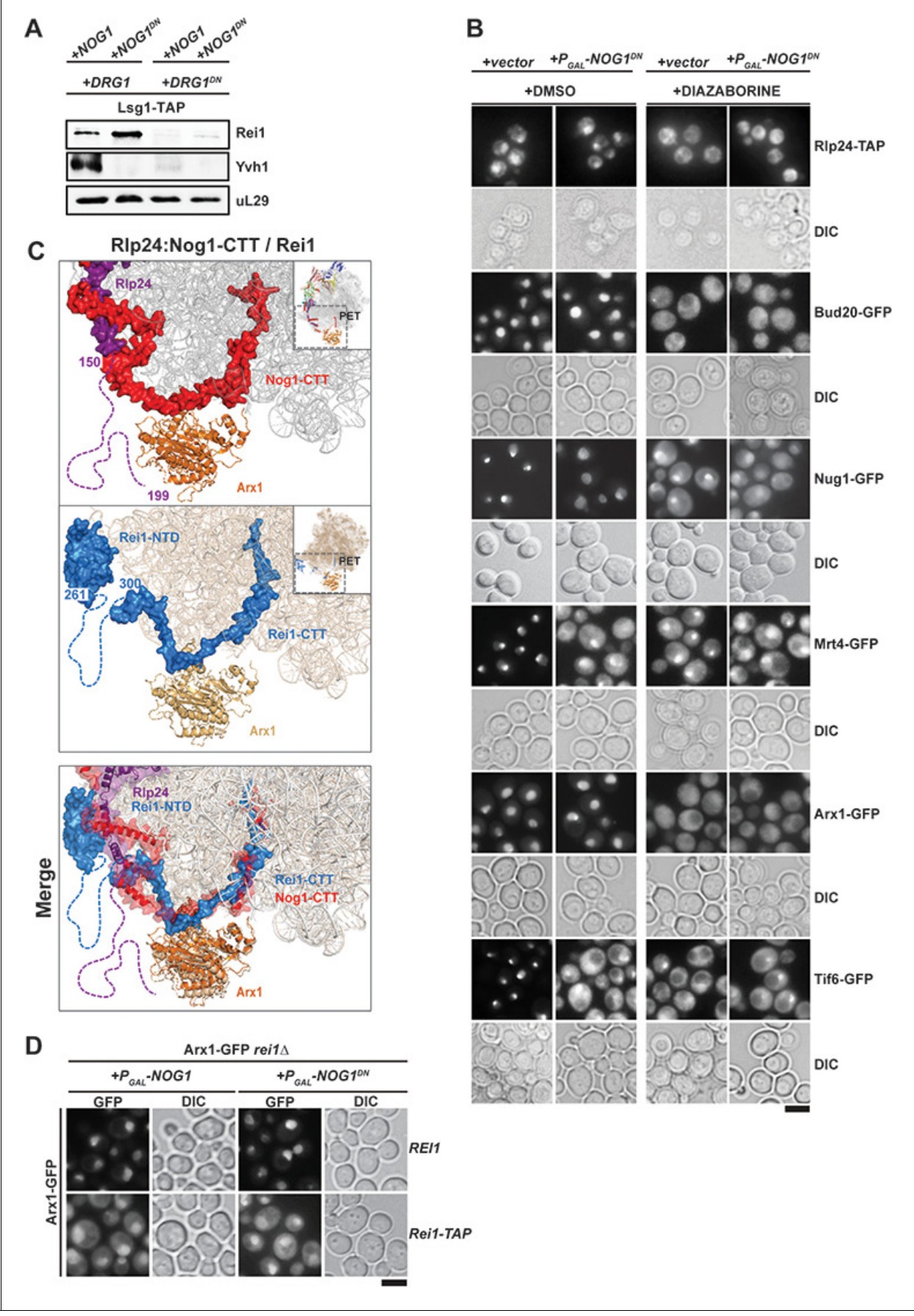

**Figure 5.** Nog1$^{DN}$ does not impair PET maturation and quality control. (A) Drg1$^{DN}$ impairs Yvh1 and Rei1 recruitment, whereas Nog1$^{DN}$ impairs only Yvh1 and not Rei1 recruitment. Lsg1-TAP was co-transformed with plasmids each expressing copper-inducible Drg1 or Drg1$^{DN}$ and galactose-inducible Nog1 or Nog1$^{DN}$, respectively. Cells were grown in raffinose-containing synthetic medium to early log phase and then supplemented with 0.5 mM copper sulphate and 2% galactose. After 3 hr, cells were lysed, and the Lsg1-TAP particle was isolated and subjected to Western analyses using indicated antibodies. (B) Indicated strains expressing Nog1$^{DN}$ under the control of GAL1 were treated with DMSO or diazaborine to block Drg1 activity and visualized by fluorescence microscopy as in *Figure 4*. Scale bar = 5 μm. Quantification of the data is listed in Supplementary Table 2. (C) The Rei1-binding site on the pre-60S clashes with the Nog1-CTT:Rlp24 complex. Cryo-EM structures of Nog1-containing (PDB-3JCT) pre-60S and a Rei1-containing (PDB-6RZZ) pre-60S were superimposed. Release of Rlp24 allows recruitment of Rei1 to the pre-60S. (D) Arx1 mislocalizes to the cytoplasm in *a rei1-TAP* mutant expressing NOG1$^{DN}$. The ARX1-GFP rei1Δ strain was transformed with either REI1- or rei1-TAP-encoding plasmids under control of their natural promoter. Subsequently, the resultant strains were transformed with plasmids encoding galactose-inducible NOG1 or NOG1$^{DN}$, respectively. The transformants were grown in raffinose-containing medium to early log phase and then supplemented with 2% galactose. After 3 hr, cells were visualized by fluorescence microscopy. Scale bar = 5 μm. Quantifications are listed in *Supplementary file 3*.

PET, and this does not require a functional Nog1-G3 domain. We conclude that the presence of Nog1$^{DN}$ on the pre-60S does not hinder Rei1 recruitment or the insertion of the Rei1-CTT into the PET.

## The Nog1-CTT:Rlp24 complex negatively regulates PET maturation

Nog1-CTT intertwines around the helical segments of Rlp24, then contacts Arx1 and finally inserts its terminal end into the PET (*Figure 1A*). We investigated which regions of Nog1-CTT contribute to pre-60S cytoplasmic maturation (*Figure 6A* and *Figure 6B*). We found that a Nog1 mutant lacking the entire CTT, including the Rlp24-interacting region, Nog1$^{1–426}$, was lethal in yeast (*Figure 6B*). Overexpression of a Nog1 mutant lacking the Rlp24-interacting region, Nog1$^{Δ427–536}$, severely impaired the growth of yeast in a dominant-negative manner (*Figure 6C*). However, unlike the Nog1$^{DN}$ mutant, Nog1$^{Δ427–536}$-expressing cells did not mislocalize Mrt4-GFP and Tif6-GFP to the cytoplasm (*Figure 6C*). Likewise, a Rlp24 mutant lacking Nog1-interacting helices, Rlp24$^{Δ91–105}$, also impaired yeast growth in a dominant-negative manner (*Figure 6A* and *Figure 6C*). Expression of the Rlp24$^{1–146}$ mutant (*Figure 6A*) that lacks the Drg1-binding platform, and therefore cannot be extracted from pre-60S, mislocalized Arx1-GFP, Mrt4-GFP and Tif6-GFP to the cytoplasm (*Figure 6C*) (*Lo et al., 2010*). However, Rlp24$^{Δ91–105}$-expressing cells did not mislocalize these factors to the cytoplasm (*Figure 6C*). Given that Nog1 and Rlp24 rely on each other for their recruitment to the pre-60S (*Saveanu et al., 2003*), the toxicity of Nog1$^{Δ427–536}$ and Rlp24$^{Δ91–105}$ mutants probably reflects a failure to form a Nog1-CTT:Rlp24 complex on the pre-60S during early nuclear maturation. In contrast to the lethal Nog1$^{1–426}$ mutant, yeast strains expressing Nog1-CTT truncation mutants containing the Rlp24-interacting region but lacking the rest of the CTT complemented the lethality of the *nog1Δ* strain, but induced impaired growth at 20°C and 25°C (*Figure 6A* and *Figure 6B*). In agreement with these observations, growth of the Nog1-GFP strain was severely impaired at these temperatures (*Figure 6B*). We monitored PET maturation (Arx1-GFP), stalk assembly (Mrt4-GFP) and Tif6-GFP localization after treating the Nog1$^{1–479}$ truncation mutant (which retains the Rlp24-interaction platform) with DIA, which impairs Drg1-mediated Rlp24 release. We found that Arx1-GFP mislocalized to the cytoplasm in these cells (*Figure 6D*), emphasizing the importance of Rlp24 release and of the disassembly of the Nog1-CTT:Rlp24 complex for PET maturation. As expected, Mrt4-GFP and Tif6-GFP also mislocalized to the cytoplasm (*Figure 6D*), consistent with the idea that Nog1 release from the pre-60S requires Rlp24 removal. In conjunction with cryo-EM studies (*Figure 5C*), these data support the idea that the Nog1-CTT:Rlp24 complex negatively regulates PET maturation.

## Nog1$^{DN}$ blocks the terminal cytoplasmic maturation steps

The terminal steps during cytoplasmic maturation of a pre-60S involve sequential release of the nuclear export signal (NES)-containing Crm1 adaptor Nmd3 and the 60S anti-association factor Tif6 (*Weis et al., 2015*; *Ma et al., 2017*; *Kargas et al., 2019*). Nog1$^{DN}$ expression accumulated Nmd3 and Tif6 on the pre-60S as determined by SWATH-MS and Western analyses (*Figure 3B* and *Figure 3C*). To monitor Nmd3 accumulation on a cytoplasmic pre-60S by cell-biological means, we employed the nuclear-localized Nmd3$^{3A}$-GFP mutant fusion, which harbors mutations in its NES that decrease its nuclear export rate (*Hedges et al., 2005*). As expected, at steady state, Nmd3$^{3A}$-GFP

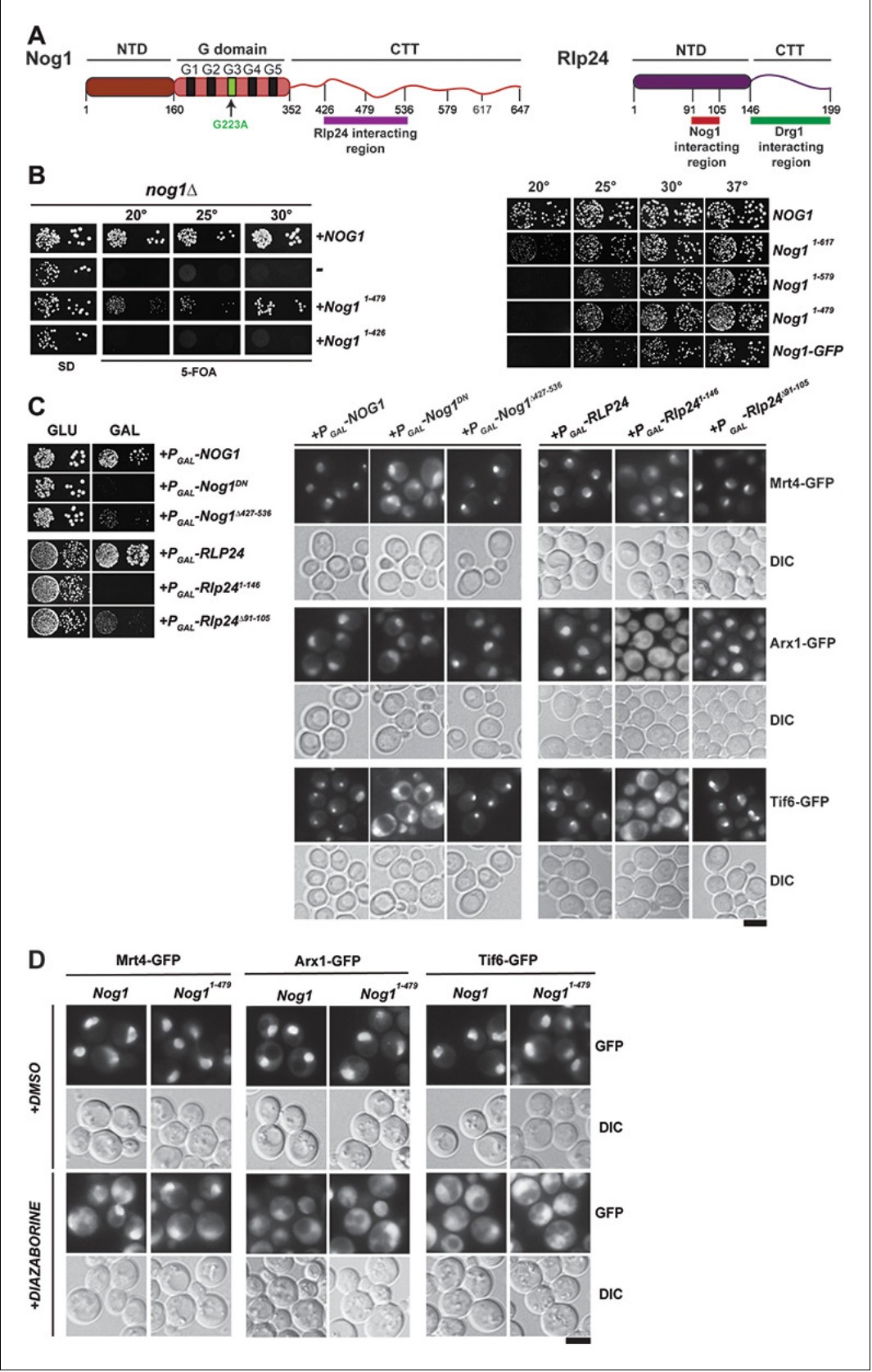

**Figure 6.** The Nog1-CTT:Rlp24 complex negatively regulates PET maturation. (**A**) Domain organization of Nog1 and Rlp24 showing interaction surfaces, and sites for truncation analyses. (**B**) Growth analysis of Nog1 truncations. *Figure 6 continued on next page*

*Figure 6 continued*

*Left panel:* a Nog1 shuffle strain was transformed with the indicated constructs and spotted in 10-fold dilutions 5'-fluoroorotic acid (5-FOA) (SD) plates and incubated for 2–6 days at the indicated temperatures. *Right panel:* a nog1Δ shuffle strain was transformed with the indicated plasmids. After shuffling out the *URA3-NOG1* plasmid on FOA-containing plates, strains were spotted in 10-fold serial dilutions on YPD plates and incubated for 2–4 days at the indicated temperatures. (**C**) Rlp24:Nog1 interactions contribute to early nucleolar/nuclear maturation. *Left panel:* BY wild-type cells were transformed with the indicated plasmids under control of the galactose-inducible GAL1 promoter and spotted in 10-fold dilutions on glucose- or galactose-containing media. Plates were incubated at 30°C for 2–4 days. *Right panel:* Mrt4-GFP, Arx1-GFP and Tif6-GFP strains were transformed with the indicated plasmids under control of the GAL1 promoter. Cells were grown in raffinose-containing synthetic medium to early log phase and then supplemented with 2% galactose. After 3 hr, cells were visualized by fluorescence microscopy. Scale bar = 5 μm. Quantifications are listed in *Supplementary file 3*. (**D**) Drg1-mediated release of Rlp24 from Nog1 is required for PET assembly. Nog1 shuffle strains containing GFP-tagged versions of assembly factors were transformed with Nog1 and Nog1$^{1-479}$ encoding plasmids. After shuffling out *URA3-NOG1* plasmid on 5-FOA, the resultant strains were grown in YPD and were treated with DMSO or diazaborine (DIA) for 30 mins to inhibit Drg1-ATPase activity. The GFP tagged assembly factor was visualized by fluorescence microscopy. Scale bar = 5 μm. Quantifications are listed in *Supplementary file 3*.

and Tif6-GFP localize to the nucleus in cells expressing WT Nog1 (*Figure 4C*). However, in Nog1$^{DN}$-expressing cells, Nmd3$^{3A}$-GFP and Tif6-GFP mislocalized to the cytoplasm (*Figure 4C*). We suggest that the presence of Nog1$^{DN}$ on the pre-60S impairs Nmd3 and Tif6 release.

Like Nog1$^{DN}$-expressing cells, *yvh1Δ* cells mislocalize both Mrt4-GFP and Tif6-GFP to the cytoplasm, impairing stalk assembly and progression of the cytoplasmic maturation pathway (*Kemmler et al., 2009*; *Lo et al., 2010*). Given that Yvh1 recruitment to the pre-60S is impaired in Nog1$^{DN}$-expressing cells, we investigated whether the dominant-negative behavior of Nog1$^{DN}$ is due to the failure to release Mrt4. To test this, we employed a dominant gain-of-function allele *MRT4$^{G68E}$* that bypasses Yvh1-mediated Mrt4 release from the pre-60S (*Figure 7A* and *Figure 7B*) and relocalizes Tif6-GFP to the nucleus (*Figure 7A*) (*Kemmler et al., 2009*). We found that Nog1$^{DN}$ expression is toxic in *yvh1ΔMRT4$^{G68E}$* cells (*Figure 7B*). Although Mrt4$^{G68E}$-GFP relocalized to the nucleus, Tif6-GFP still mislocalized to the cytoplasm in these cells (*Figure 7A*), showing that the presence of Mrt4 on the pre-60S is not the cause of the dominant-negative behavior of Nog1$^{DN}$. Consistent with the sequential release of Nmd3 and Tif6 from the pre-60S, Western analyses show that Nmd3 accumulates on the Lsg1-TAP particle isolated from *yvh1ΔMRT4$^{G68E}$* cells expressing Nog1$^{DN}$ (*Figure 7C*). Notably, the r-protein uL10 is recruited to this Lsg1-TAP particle (*Figure 7C*), indicating that Nog1$^{DN}$ does not interfere with stalk assembly in *yvh1ΔMRT4$^{G68E}$* cells. In light of a recent cryo-EM study, these data suggest that Nog1$^{DN}$ presence on the pre-60S prevents the rearrangement of rRNA helix H89, and sterically hinders the incorporation of uL16 into its RNA-binding site (*Figure 7D*), a critical step that triggers downstream events that drive a pre-60S toward translation competence (*Kargas et al., 2019*; *Zhou et al., 2019*).

## Discussion

During its journey from the nucleolus to the cytoplasm, and concomitant progression towards the mature subunit, a pre-60S interacts with diverse energy-consuming enzymes. However, the precise order of recruitment and eviction, as well as the mechanisms that co-ordinate the different energy-consuming steps remain unknown. Here, we reveal that sequential Drg1-ATPase and Nog1-GTPase activities license early cytoplasmic maturation events that eventually drive a pre-60S towards translation competence.

### Nog1 eviction requires a functional G-domain and Drg1-ATPase activity

The GTPase Nog1 travels into the cytoplasm with the pre-60S, where it is released in a Drg1-ATPase manner and recycled back into the nucleus (*Pertschy et al., 2007*; *Lo et al., 2010*). Our finding that the dominant negative Nog1$^{DN}$ (Nog1G223A) mutant accumulates on Lsg1-TAP (*Figure 1C*) supports the notion that an impaired G-domain does not hinder recruitment to the assembling pre-60S particle. Instead, like other GTPases involved in the 60S maturation pathway (*Hedges et al., 2005*; *Matsuo et al., 2014*), an impaired G3 domain blocks Nog1 release from the pre-60S. We and others

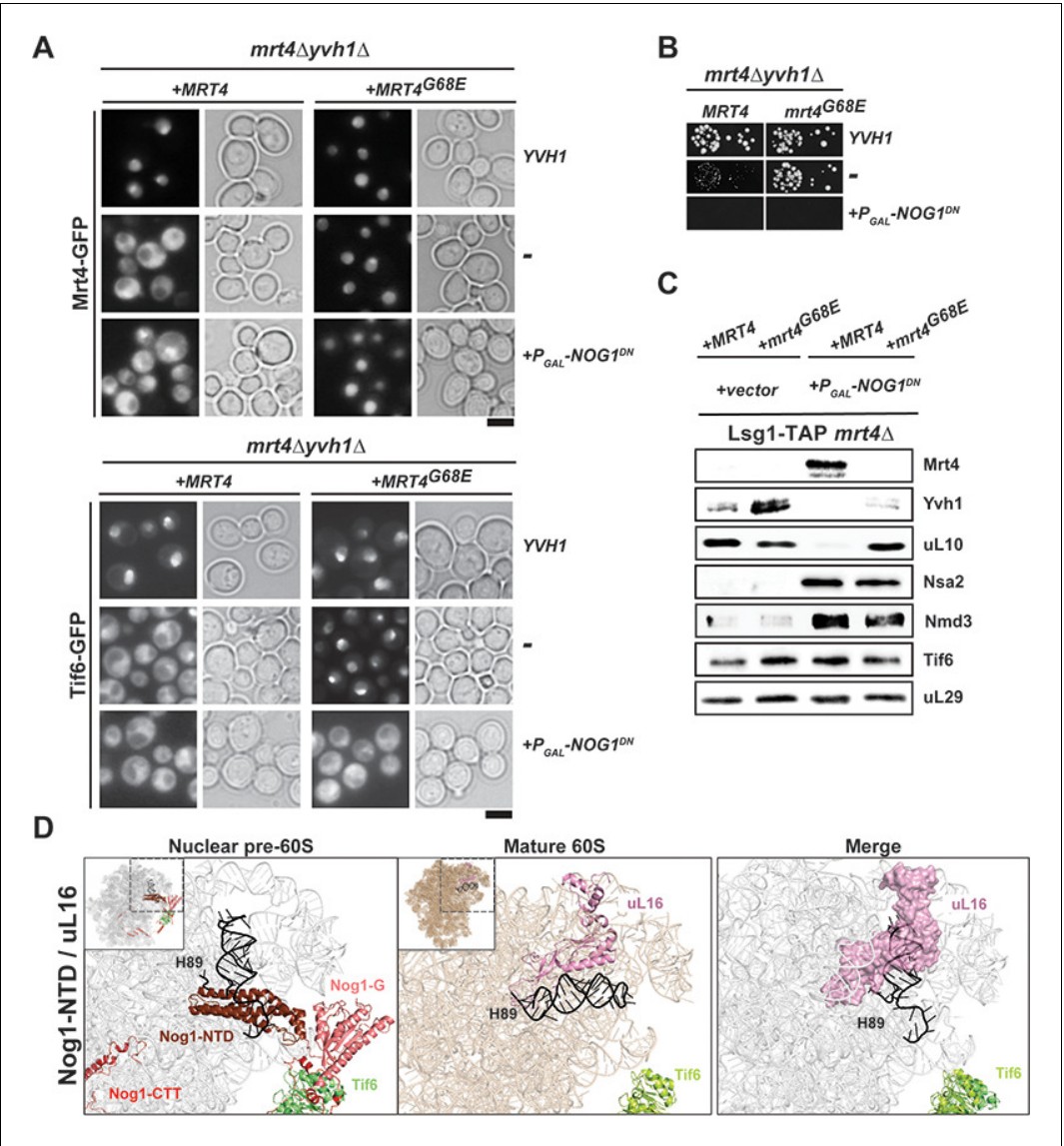

**Figure 7.** Nog1[DN] blocks the terminal cytoplasmic maturation steps. (**A**) Nog1[DN] impairs recycling of Mrt4 and Tif6. The *mrt4Δyvh1Δ* strain was co-transformed with plasmids encoding Mrt4-GFP or Mrt4G68E-GFP and Yvh1 or galactose-inducible Nog1[DN], respectively. To monitor Tif6 localization, the strain was co-transformed with plasmids encoding Tif6-GFP, Mrt4 or Mrt4G68E-GFP and Yvh1 or galactose-inducible Nog1[DN], respectively. Cells were grown in raffinose-containing synthetic medium to early log phase and then supplemented with 2% galactose. After 3 hr, cells were visualized by fluorescence microscopy. Scale bar = 5 μm. Quantifications are listed in **Supplementary file 3**. (**B**) Mrt4G68E bypasses the need for *YVH1*, but does not rescue lethality associated with Nog1[DN] expression. The depicted strains were spotted in 10-fold dilutions on galactose-containing media and incubated at 30°C for 2–4 days. (**C**) Nog1[DN] impairs recruitment of Yvh1, but not release of the Mrt4-G68E gain-of-function mutant. A Lsg1-TAP *mrt4Δ* strain was co-transformed with plasmids encoding Mrt4 or Mrt4G68E and empty vector or galactose-inducible Nog1[DN], respectively. Cells were grown in raffinose-containing synthetic medium to early log phase and then supplemented with 2% galactose. After 3 hr, cells were lysed, purified through Lsg1-TAP and subjected to western analyses using the indicated antibodies. (**D**) Nog1 release permits uL16 recruitment. Superimposition of the cryo-EM structures of a nuclear Nog1-containing pre-60S particle (PDB-3JCT) and uL16 on a mature 60S subunit (PDB-6R84) show how the Nog1-NTD and Nog1 G-domain prevent recruitment of uL16 to the pre-60S.

(**Lapik et al., 2007**) have been unable to detect GTPase activity of recombinant Nog1 in vitro. It could be that GTP hydrolysis of Nog1, like that of its bacterial homolog ObgE, is triggered only in

the context of the pre-ribosome (*Feng et al., 2014*). Nog1 is a predicted potassium-selective cation-dependent GTPase (also referred to as Group I CD-GTPase) (*Ash et al., 2012*). Group I CD-GTPases can be identified by the presence of two conserved asparagine residues in the G1 motif (*Ash et al., 2012*). The first of the two conserved asparagine residues (Asn$^K$) is coordinated to the potassium ion, whereas the second asparagine (Asn$^{SwI}$) facilitates the Switch I structure. In yeast Nog1, Asn$^K$ is present (N177), however, the asparagine Asn$^{SwI}$ is substituted with an arginine (R185). This raises the possibility that the second asparagine (or equivalent) might be provided by a yet unknown activator. The equivalent G223A mutation in the G3 motif reported here is also dominant negative in the Ras GTPase (*Ford et al., 2005*). Structural studies of this mutant revealed that the switch region adopts an open conformation that prevents GTP from inducing an active confirmation (*Ford et al., 2005*), which sequesters Ras GEFs into non-productive complexes leading to the depletion of their intracellular pool (*Ford et al., 2005*). It would be interesting to determine the molecular basis that underlies the dominant-negative nature of Nog1$^{DN}$, and to determine whether it is similar to that for the Ras$^{G60A}$ mutant.

Nog1, whose GTPase domain is docked at the P0 stalk base at a position similar to that in bacterial ObgE, is remarkably unique with its CTT, which wraps around half the 60S subunit, passing by the PTC and entering the PET (*Wu et al., 2016*) (*Figure 1A*). Covering a total distance of >250 Å, the CTT contacts Rlp24, Bud20, Tif6 and Arx1 (*Figure 1A*). Nog1-CTT intertwines with the C-terminal helices of Rlp24 forming a Nog1-CTT:Rlp24 complex on the pre-60S (*Figure 4A*). Intriguingly, despite these intimate interactions, we found that Rlp24 and Nog1$^{DN}$ are independently released from the pre-60S. Previous work showed that Drg1 grips Rlp24 via its C-terminus, and then triggers its ATP-dependent release from the pre-60S (*Kappel et al., 2012*). Drg1$^{DN}$ expression prevents the release of both Rlp24 and Nog1 from the pre-60S, suggesting that Nog1 eviction also requires Drg1-ATPase activity (*Pertschy et al., 2007*; *Altvater et al., 2012*). A recent cryo-EM study identified an early cytoplasmic pre-60S state in which Nog1-CTT was inserted into the PET, but the Nog1-NTD and Nog1-G domains were displaced from the pre-60S (*Zhou et al., 2019*). In this particle, Nog1-CTT intertwines around Rlp24, providing an explanation as to why Nog1 remains bound to the pre-60S. We suggest that Nog1 eviction from the pre-60S requires a functional Nog1 G-domain and Drg1-mediated extraction of Rlp24.

Another factor that makes contact with Rlp24 is the export factor Bud20 (*Altvater et al., 2012*; *Bassler et al., 2012*). It seems likely that Bud20 release from the pre-60S is a consequence of Rlp24 release. Consistent with this, Drg1$^{DN}$ expression interferes with both Rlp24 and Bud20 release and recycling from the pre-60S. Given that localization of Rlp24 and Bud20 is not altered upon Nog1$^{DN}$ expression (*Figure 4C*), we suggest that Bud20 release from the pre-60S occurs prior to bifurcation of the cytoplasmic maturation pathway.

## The Nog1-CTT:Rlp24 complex gates PET quality control

Drg1$^{DN}$ expression impairs Rei1 recruitment to the pre-60S and consequently Arx1 release (*Altvater et al., 2012*). Drg1-mediated Rlp24 extraction from the Nog1-CTT:Rlp24 complex permits Rei1 recruitment to the pre-60S, and subsequently allows the Rei1-CTT to probe PET integrity for Arx1 release. Rei1 recruitment to the pre-60S is unaffected in Nog1$^{DN}$-expressing cells. Given that PET maturation is successfully accomplished in Nog1$^{DN}$-expressing cells, we suggest that Nog1$^{DN}$-CTT has been extracted from the PET as a consequence of Rlp24-extraction from the pre-60S. Thus, Nog1 eviction from the pre-60S is not required to accomplish PET maturation.

Recent cryo-EM studies revealed that the cytoplasm-localized assembly factor Reh1, which functionally overlaps with Rei1 (*Parnell and Bass, 2009*; *Greber et al., 2016*) also inserts its C-terminal tail into the PET (*Ma et al., 2017*; *Kargas et al., 2019*). Our SWATH-MS data indicate that Reh1 accumulates on the Lsg1-TAP:Nog1$^{DN}$ particle (*Figure 3B*), supporting the notion that Reh1 release occurs after Nog1 release from the pre-60S. These data are consistent with the idea that the PET is under continuous surveillance through sequential interactions with Nog1-CTT, Rei1-CTT and Reh1-CTT.

## Nog1 gates stalk assembly and PTC maturation

Stalk assembly requires prior release of Mrt4 that is facilitated by the release factor Yvh1 (*Lo et al., 2009*; *Kemmler et al., 2009*). Nog1$^{DN}$ expression interferes with Yvh1 recruitment to the pre-60S

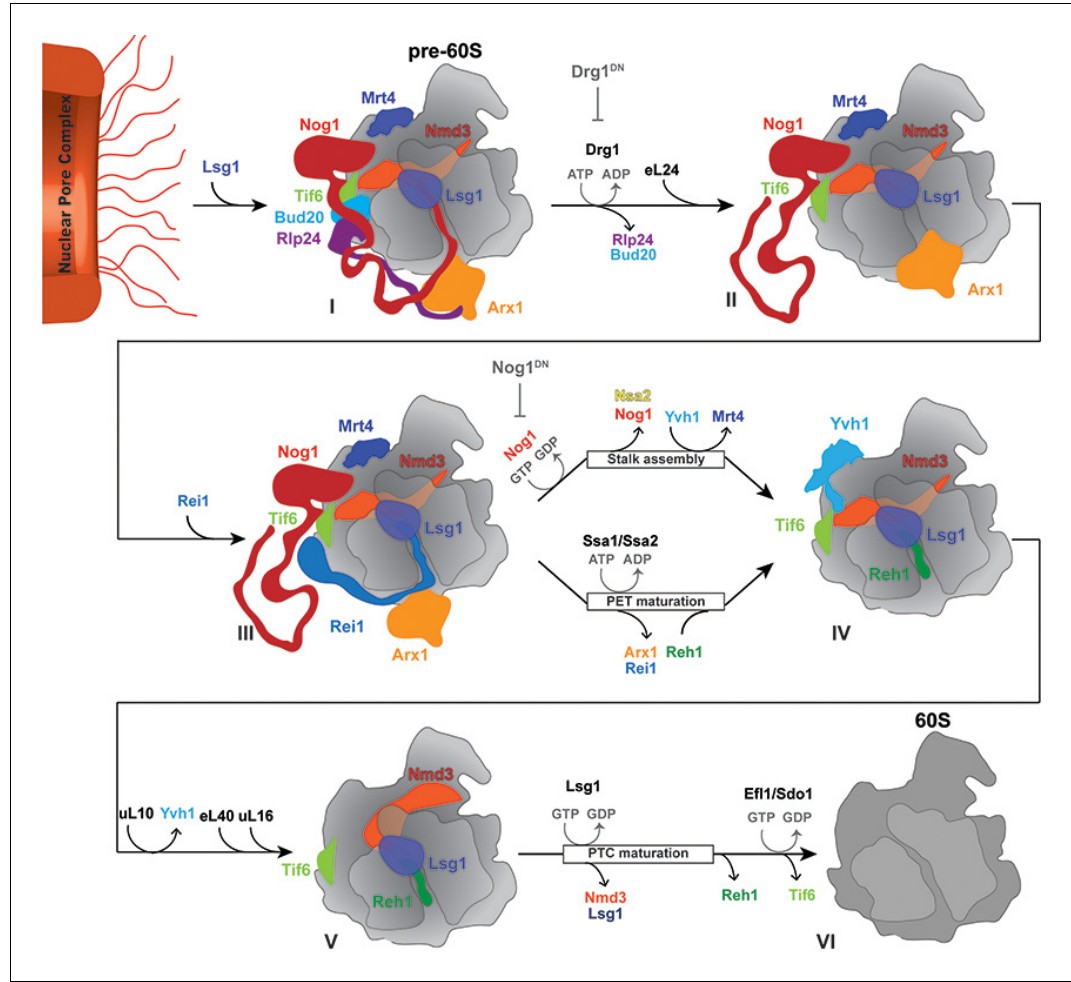

**Figure 8.** A revised model for cytoplasmic maturation of the pre-60S. Drg1 initiates the 60S cytoplasmic maturation pathway by binding to and releasing Rlp24 from the pre-60S bound Rlp24:Nog1-CTT complex in an ATP-dependent manner (State I to State II). Removal of Rlp24 also releases its interacting partner Bud20, and consequently extracts the Nog1 C-terminal tail out of the polypeptide exit tunnel (PET) (State II). Extraction of Nog1-CTT from the PET, permits Rei1-CTT to probe PET integrity (State III). Successful Rei1-CTT insertion into the PET triggers ATP-dependent release of Arx1 by Ssa1/2 and Jjj1, and concurrent Rei1 eviction (State III to State IV). This permits Reh1-CTT insertion into the PET (State IV). Nog1 release permits Yvh1 recruitment to evict Mrt4 and to initiate stalk assembly (State IV to State V). Nog1 eviction also permits eL40 and uL16 incorporation, and Nsa2 release. Stable uL16 accommodation into the pre-60S leads to Lsg1 mediated Nmd3 release, and thereby completes PET maturation. Nmd3 release triggers Reh1 release, and also licenses the GTPase Efl1 and co-factor Sdo1 to evict Tif6 (State V to State VI), thus driving a pre-60S towards translation competence.

particle, and therefore impairs Mrt4 release (*Figure 4C* and *Figure 4D*). Cryo-EM studies have revealed that the presence of Nog1 prevents the pre-60S from recruiting Yvh1, as both factors localize close to the stalk base and obstruct the Yvh1-binding site (*Figure 4D*) (*Kargas et al., 2019*; *Zhou et al., 2019*). Given that Nog1 release is impaired in Drg1[DN]-expressing cells (*Altvater et al., 2012*), these data provide an explanation as to why stalk assembly is impaired in these cells.

While Mrt4[G68E] efficiently bypassed the need for Yvh1-mediated release, this gain-of-function allele did not rescue the dominant-negative phenotype of Nog1[DN] (*Figure 7B*). These cells were still impaired in Nmd3 and Tif6 release from the pre-60S (*Figure 7A* and *Figure 7C*). Retention of Nog1[DN] on the pre-60S prevents rearrangement of rRNA helix H89, and sterically hinders uL16 incorporation. Stable accommodation of uL16 into the pre-60S is a critical event to re-orient rRNA helices H89 and H38, that permits a conformational rearrangement of the Nmd3 C-terminal domain (*Malyutin et al., 2017*; *Zhou et al., 2019*; *Kargas et al., 2019*). This rearrangement triggers the

GTPase Lsg1 to evict Nmd3, and subsequently licenses Efl1/Sdo1 to mediate Tif6 release. Notably, the dedicated chaperone Sqt1 that guides incorporation of uL16 into the pre-60S accumulates on the Lsg1-TAP:Nog1[DN] particle (*Figure 3B*). In light of these data and work from the Kressler laboratory (*Pausch et al., 2015*), it is tempting to speculate that a pre-formed Sqt1:uL16 complex docks on the pre-60S. Nog1 eviction permits transfer of uL16 into its RNA-binding site, and release of its dedicated chaperone Sqt1. We suggest that Nog1 release from a pre-60S licenses stalk assembly and downstream maturation steps that are critical to acquire translation competence.

SWATH-MS analyses allowed the identification of assembly factors whose release or recruitment appears to be dependent on the release of the GTPase Nog1 from cytoplasmic pre-60S particles. These include the assembly factors Sqt1, Mex67, Ecm1, Tma16 and Arb1, for which structural information in the context of a pre-60S is lacking. In contrast to Sqt1, Mex67, Ecm1 and Tma16, which accumulated on the Lsg1-TAP Nog1[DN] particle, we found that recruitment of the ABC-ATPase Arb1 was impaired (*Figure 3B*). Arb1 has been implicated in both 40S and 60S subunit assembly (*Dong et al., 2005*). A recent cryo-EM structure of a translationally stalled-mature 60S subunit shows that Arb1 docks in a region of rRNA that overlaps with the binding site of Nmd3 on the pre-60S (*Su et al., 2019*). We speculate that Nog1 release, stable uL16 incorporation and Nmd3 release precede Arb1 recruitment to the pre-60S.

## Nog1 co-ordinates the assembly, maturation and quality control of distant ribosomal centers

Previously, the Johnson laboratory had ordered cytoplasmic maturation events on the pre-60S into a coherent pathway (*Lo et al., 2010*). In that pioneering study, AAA-ATPase Drg1 activity was proposed to initiate two parallel branches of the pathway, namely PET maturation with quality control and ribosome stalk assembly (*Figure 3A*). The two branches converge to initiate Nmd3 and Tif6 release prior to acquiring translational competence. Our data suggest that Nog1 plays a pivotal role in co-ordinating bifurcation and convergence of the 60S cytoplasmic maturation pathway. Cryo-EM structures and the data presented here permit the refinement of early events that drive the cytoplasmic pre-60S maturation (*Figure 8*). Upon arrival in the cytoplasm, the pre-60S recruits Drg1 via the C-terminal region of Rlp24. This interaction stimulates Drg1 ATPase activity resulting in Rlp24 removal, and as a consequence, the terminal end of Nog1-CTT is extracted from the PET (State I to State II). Removal of Rlp24 permits Rei1 recruitment, and allows Rei1-CTT to probe PET integrity in order to trigger Arx1 release through the J-domain protein Jjj1 and ATPases Ssa1/2 (State III) (*Meyer et al., 2007*; *Greber et al., 2016*). Release of Rei1 then permits the assembly factor Reh1 to probe the PET (State IV) (*Ma et al., 2017*; *Kargas et al., 2019*). Nog1 eviction from the pre-60S requires the extraction of Rlp24 from Nog1-CTT by the ATPase Drg1 and GTP hydrolysis within the Nog1-G domain. This event unleashes a sequence of events in different regions of the pre-60S: (1) it permits recruitment of the release factor Yvh1 to the pre-60S in order to mediate Mrt4 release and initiate stalk assembly (State IV to State V); (2) it permits loading of r-proteins eL40 and uL16 into their rRNA-binding sites (State V), (*Zhou et al., 2019*; *Kargas et al., 2019*); and (3) it re-orientates rRNA helices H89 and H38 to promote a conformational rearrangement of the Nmd3 C-terminal domain (*Malyutin et al., 2017*; *Zhou et al., 2019*; *Kargas et al., 2019*), leading to Nmd3 eviction and thus completing PTC maturation (State V to State VI). We therefore propose that Nog1 serves as a hub that coordinates spatially distant maturation events to ensure only a correctly assembled 60S subunit enters translation.

Finally, Nsa2 and Reh1 accumulate on Lsg1-TAP in Drg1[DN]- as well as Nog1[DN]- expressing cells (*Altvater et al., 2012*; *Figure 3C*). These data are consistent with the idea that Nog1 eviction from the pre-60S precedes Nsa2 and Reh1 release. However, the precise mechanisms that drive Nsa2 and Reh1 release remain unknown.

Nmd3 release licenses Efl1/Sdo1 to mediate Tif6 release (State VI) (*Kargas et al., 2019*). Although Lsg1 appears to be required only during a specific step of the cytoplasmic maturation pathway, it seems that it can be recruited to a newly exported pre-60S. Lsg1 recruitment to the pre-60S does not seem to depend on the initial cytoplasmic maturation events, such as Rlp24 extraction by Drg1 or Nog1 release, as we efficiently isolated Drg1[DN]- and Nog1[DN]-trapped particles via Lsg1-TAP (*Altvater et al., 2012*) (*Figure 3B*). These purifications suggest that the GTPases Nog1 and Lsg1 can bind simultaneously to cytoplasmic pre-60S particles.

## Co-ordination between energy-consuming enzymes ?

Strikingly, the release of the GTPase Nug2 from the pre-60S in the nucleus was reported to require its own GTPase activity, and the ATPase activity of the AAA-ATPase Rea1 (*Matsuo et al., 2014*). Nug2 binds the pre-60S at a site, which partially overlaps with the binding site for the NES-containing adaptor Nmd3 (*Sengupta et al., 2010*). Only after Nug2 release can Nmd3 be recruited to the pre-60S, a critical step that drives pre-60S nuclear export. In this case, coupled ATPase and GTPase activities, together with components of this assembly factor cluster (*Figure 2*), form a nuclear checkpoint that prevents a pre-60S from prematurely acquiring export competence. Our data suggest that Nog1 release, like Nug2 release, from the pre-60S presumably requires its own GTPase activity, and Drg1 ATPase activity. Failure to release Nog1 blocks a specific branch of the cytoplasmic maturation pathway, and consequently PTC formation.

In contrast to their prokaryotic counterparts, formation of the eukaryotic ribosome requires the concerted efforts of >200 assembly factors and >40 ATPases, GTPases, ATP-dependent RNA helicases and kinases that interact with pre-ribosomes at distinct maturation steps. Our workflow of combining genetic trapping with SWATH-MS provides a powerful tool to uncover the intricate coordination between energy-consuming enzymes to release assembly factor clusters sequentially from maturing pre-ribosomes, and thereby reveal checkpoints during ribosome formation.

# Materials and methods

## Yeast strains and plasmids

The *Saccharomyces cerevisiae* strains used in this study are listed in *Supplementary file 1*. Genomic disruptions, C-terminal tagging and promoter switches at genomic loci were performed according to established protocols (*Janke et al., 2004*; *Longtine et al., 1998*; *Puig et al., 2001*).

The plasmids used in this study are listed in *Supplementary file 2*. Details of plasmid construction will be provided upon request. All recombinant DNA techniques were performed according to established procedures using *Escherichia coli* XL1 blue cells for cloning and plasmid propagation. Mutations in *NOG1* and *RLP24* were generated using the QuikChange site-directed mutagenesis kit (Agilent Technologies, Santa Clara, CA, USA). All cloned DNA fragments and mutagenized plasmids were verified by sequencing.

## Fluorescence microscopy

For assessing the localization of PGAL1-Nog1G223A-GFP, BY4741 cells were transformed with YEP351-Nog1-GFP and YEP351gal-Nog1G223A-GFP and grown in SR medium until OD600 = 0.4–0.6, then induced for 30 min with 2% galactose. Cells were then washed once in YPD and incubated in YPD for 3 hr before imaging. For assessing the localization of assembly factors, endogenously GFP-tagged strains were transformed with PGAL1 plasmids and were grown in SR medium until OD600 = 0.2–0.4, and then induced with 2% galactose for 3 hr. For Nmd3$^{3A}$-GFP visualization, a Nmd3 shuffle strain was transformed with Nmd3$^{3A}$-GFP (NMD3I493A L497A L500A) and transformants were applied on FOA plates to shuffle out the wild-type *NMD3*. The resulting Nmd3$^{3A}$-GFP strain was transformed with PGAL1 plasmids and grown the same as the endogenously GFP-tagged strains. For experiments with diazaborine, cells were incubated 1 hr prior to imaging with either DMSO or with 370 µM diazaborine (dissolved in DMSO; M. Peter, ETH Zürich, Switzerland) in 1 ml of cell culture for 1 hr at 30℃ on shaker. When cells were ready to be harvested, the pellet was washed once with H$_2$O. 3 µl of cells were transferred on a microscopy slide (VWR) covered with a glass slip (18 × 18 mm no 1, VWR).

Rlp24-TAP localization was assayed by indirect immunofluorescence. The Rlp24-TAP strain was transformed with pGAL1-containing plasmids, and cultures were grown in the appropriate conditions to OD600 = 0.4–1. After adding formaldehyde (final concentration 4%) to the cultures, cells were fixed for 30 min at 30℃. Cells were then centrifuged, and incubated with 1 ml of 0.1M KPi (pH 6.4) with 3.7% formaldehyde for 15 min at 30℃. Cells were then washed twice with 0.1M KPi (pH 6.4) and once in spheroplasting buffer (0.1 M KPi [pH7.4], 1.2 M sorbitol, 0.5 mM MgCl$_2$). The pellet was resuspended in 200 µl spheroplasting buffer and was either stored at −20℃ or directly processed for spheroplasting. For spheroplasting, 2 µl of 1M DTT was added to the 200 µl cells and incubated for 15 min at 30℃. Zymolyase 100T was added (final concentration 50 µg/ml) to the 200 µl

and incubated for 10 min. Cells were quickly checked under the microscope to see whether sphero-plasts were formed (spheroplasts appear black), and if necessary incubated for a longer time, but no longer than 20 min. Then, the spheroplasts were centrifuged for 2 min at 2000 rpm, washed and resuspended in spheroplasting buffer. 20 µl of poly-L-lysine (0.1% [w/v], Sigma-Aldrich) was applied per well on slides (8-well, 6-mm Menzel-Gläser Diagnostika, Brauschweig, Germany), incubated for 5 min, washed three times with dH$_2$O, aspirated and air-dried. Approximately 40 µl of spheroplasts were added onto a lysine-coated slide well. Non-adhering cells were removed after 30 s and the slide was incubated in an ice-cold methanol bath for 6 min before it was transferred to an ice-cold acetone bath for 10 s. The slide was air-dried and processed with 30 µl BSA/PBS (1x PBS, 1% w/v BSA) for 30 min in the dark at room temperature (RT) in a humid chamber. The blocking solution was removed, and primary antibody (anti-CBP 1:1000; Thermo Scientific, Rockford, IL, USA) in BSA/PBS was added to the wells and incubated for at least an hour (up to overnight) at RT in the dark. Wells were washed three times with BSA/PBS and incubated with secondary antibody, Alexa-Fluor568 coupled anti-rabbit (Molecular Probes, Inc, Eugene, OR, US), at RT in the dark. After wash-ing, cells were incubated for 30 s with DAPI (1 µg/ml in BSA/PBS; Sigma-Aldrich). Wells were washed three times with BSA/PBS and dried before the slides were mounted with Mowiol (Calbio-chem, San Diego, CA, USA).

Fluorescence signal was examined using a Leica DM6000B microscope fitted with a 63 × 1.25 NA 1.30–0.60 NA oil immersion lens (HCX PL Fluotar; Leica). Pictures were acquired with a digital cam-era (ORCA-ER; Hamamatsu Photonics) and with Openlab software (Perkin Elmer) or with Leica LAS software. Representative sections were selected using ImageJ software and processed in AdobePho-toshop. Corresponding differential interference contrast (DIC) pictures were taken for each fluores-cence image. All cell-biological studies were performed on at least three different occasions and in triplicates; >90% of cells showed the reported phenotypes in a sample size of >1000 cells. A sum-mary of the quantification of all of the cell-biological data is listed in *Supplementary file 3*.

## Biochemical analyses

Whole-cell extracts were prepared by alkaline lysis of yeast cells (*Kemmler et al., 2009*). Tandem affinity purifications (TAP) of pre-ribosomal particles were carried out as previously described (*Faza et al., 2012*; *Altvater et al., 2014*). Calmodulin-eluates were separated on NuPAGE 4–12% Bis-Tris gradient gels (Invitrogen, Carlsbad, CA, USA) and visualized by either silver staining or west-ern analyses using indicated antibodies. All biochemical purifications were performed on at least three different occasions and in triplicates. To analyze the samples by SWATH-MS, protein TEV eluates were precipitated using trichloroacetic acid (TCA), washed once in cold acetone, air dried and processed further (see paragraphs on SWATH-MS analyses).

Western analyses were performed as previously described (*Kemmler et al., 2009*). The following antibodies were used: α-Mex67 (1:5000; C Dargemont, Institut Jacques Monod, Paris, France), α-Nmd3 (1:5000; A Johnson, University of Texas at Austin, Austin, TX, USA), α-Nog1 (1:1000; M Fro-mont-Racine, Institut Pasteur, Paris, France), α-Nop7 (1:2000; B Stillman, Cold Spring Harbor Labora-tory, New York, NY, USA), α-Nug1 (1:1000; this study), France), α-Tif6 (1:2000; GenWay Biotech, San Diego, CA, USA), α-Yvh1 (1:4000; *Altvater et al., 2012*), α-Rpl1 (1:10,000; F Lacroute, Centre de Génétique Moléculaire du CNRS, Gif-sur-Yvette, France), α-Rpl3 (1:5000; J Warner, Albert Einstein College of Medicine, Bronx, NY, USA), α-Rpl35 (uL29) (1:4000; *Altvater et al., 2012*), α-Gsp1 (1:3000; rabbit; *Fischer et al., 2015*), α-Nsa2 (1:2000; M. Fromont-Racine, Institut Pasteur, Paris, France), α-GFP (1:2000; Roche); and α-FLAG (1:2000; Sigma). The secondary HRP-conjugated α-rab-bit and α-mouse antibodies (Sigma-Aldrich, St. Louis, MO, USA) were used at 1:1000–1:5000 dilu-tions. Protein signals were visualized using an Immun-Star HRP chemiluminescence kit (Bio-Rad Laboratories, Hercules, CA, USA) and captured by Fuji Super RX X-ray films (Fujifilm, Tokyo, Japan) or using ImageQuant LAS 4000 (GE Healthcare).

## Sample preparation for the SWATH-MS analyses

The proteins were solubilized in a denaturing buffer containing 8 M urea and 0.1 M NH$_4$HCO$_3$. They were then reduced with 12 mM DTT at 37°C for 30 min and alkylated with 40 mM iodacetamide at room temperature in the dark for 30 min. The samples were then diluted with 0.1 M NH$_4$HCO$_3$ to reach a final urea concentration of 1M and digested with sequencing-grade porcine trypsin

(Promega, 1:100 trypsin:protein). Digestion was stopped by adding formic acid (FA) to a final concentration of 1% (pH ~2). Peptides were desalted using macro spin columns (Nest group) according to the following procedure: cartridges were wetted with one volume (350 µl) 100% methanol, washed with two volumes of 80% acetonitrile and 0.1% FA, and equilibrated with three volumes of 0.1% FA. The acidified peptides were loaded twice on the cartridge, washed with three volumes of 0.1% FA and eluted with two volumes of 50% acetonitrile and 0.1% FA. Peptides were dried in a speedvac concentrator and resolubilized in 10 µl of 0.1% FA. The samples were then transferred to an mass spectroscopy (MS) vial and spiked with 1:20 (v:v) of iRT peptides (*Escher et al., 2012*).

## Mass spectrometry data acquisition

1 µg of peptides were injected on a 5600 TripleTof mass spectrometer (ABSciex, Concord, Ontario, Canada) interfaced with an Eksigent NanoLC Ultra 1D Plus system (Eksigent, Dublin, CA, USA). The peptides were separated on a 75-µm-diameter, 20-cm long New Objective emitter packed with Magic C18 AQ 3 µm resin (Michrom BioResources) and eluted at 300 nl/min with a linear gradient of 5–35% Buffer A for 120 min (Buffer A: 2% acetonitrile, 0.1% FA; Buffer B: 98% acetonitrile, 0.1% FA). MS data acquisition was performed in either data-dependent acquisition (DDA, top20, with 20 s dynamic exclusion) or data-independent acquisition (DIA) SWATH-MS mode (32 fixed precursor isolation windows of 25 Da width [+1 Da overlap] each acquired for 100 ms plus one MS1 scan acquired for 250 ms) as described in *Gillet et al. (2012)*. The mass ranges recorded were 360–1460 m/z for MS1 and 50–2000 m/z for MS2. For either mode, the collision energy was set to $0.0625 \times$ m/z – 6.5 with a 15 eV collision energy spread regardless of the precursor charge state.

## SWATH-MS assay library generation

The DDA data recorded as described above were used to generate an assay library essentially as described (*Schubert et al., 2015*). In short, the raw DDA files were converted to mzXML using the qtofpeakpicker component of msconvert (ProteoWizard v 3.0.9987). The converted files were searched with Comet (2014.02 rev. 0) and Mascot (version 2.5.1) using the yeast SGD database (release 13.01.2015), which contained 6713 proteins plus one protein entry for the concatenated sequence of the iRT peptides and as many decoy protein entries generated by pseudo-reversing the tryptic peptide sequences. The search parameters were as follows: + /– 25 ppm tolerance for MS1 and MS2, fixed cysteine carbamidomethylation, variable methionine oxidation, semi-tryptic and up to two missed cleavages per peptide allowed. The Comet and Mascot search results were further processed using peptideProphet (*Keller et al., 2002*) and aggregated using iProphet (*Shteynberg et al., 2011*) (TPP v4.7 rev 0). The search results were filtered for an iProphet cutoff of 0.877603, corresponding to a 1% protein false discovery rate (FDR) estimated by MAYU (*Reiter et al., 2009*). The search results contained 22,961 peptides matching a set of 2610 proteins. The consensus spectral library was generated using SpectraST (*Lam et al., 2008*), and the assay library thereof was exported using the spectrast2tsv.py script (*Schubert et al., 2015*) with the following parameters: six highest intensity fragments (of charge 1+ or 2+) per peptide, within the mass range 350–2000 m/z and excluding the fragments within the precursor isolation window of the corresponding swath. The final library contains assays for 23,960 peptide precursors (including 21,472 proteotypic precursors covering 2209 unique proteins). The assay library was exported to a TraML format with shuffled decoys appended as described (*Schubert et al., 2015*).

## SWATH-MS data analysis

The SWATH-MS data were extracted with the above-mentioned assay library through the iPortal interface with openSWATH (*Röst et al., 2014*) (openMS 2.1.0), pyProphet (*Teleman et al., 2015*) and TRIC alignment (*Röst et al., 2016*) using the parameters described in *Navarro et al. (2016)*. The SWATH identification results were further filtered to keep all the peptide assays with m-score below 0.01 for the protein entries with at least one peptide with an m-score below 0.00000208508 (corresponding to a protein FDR of 1%). A set of ribosomal proteins (RPL28, RPL14B, RPL14A, RPL3, RPL19A/RPL19B, RPL35A/RPL35B, RPL12A/RPL12B, RPL27A/RPL27B, RPL2B/RPL2A, RPL14A/RPL14B, RPL7B/RPL7A, and RPL1A/RPL1B) was used to normalize (mean-center) the data as described in *Altvater et al. (2012)*. Only the proteotypic peptides were then kept, as well as the assays identified in the three triplicates for at least one AP-MS condition. Finally, proteins with fewer

than two peptides were filtered out. The missing values for each peptide assay were imputed using a random value between 0.7- and 0.9-fold the lowest intensity of that peptide assay throughout the dataset. All of the peptide assay values were then summed to a protein intensity value and a protein intensity mean for each AP-MS condition. For the general protein heatmaps, each protein intensity was normalized to that of the highest protein value for the reported set. For the bar plots, the fold changes were calculated for each protein relative to that of the AP-MS condition with the highest protein intensity, while the standard deviations were calculated for each condition.

## Acknowledgements

We thank M Peter, C Dargemont, A Johnson, M Fromont-Racine, B Stillman, F Lacroute, and J Warner for generously sharing yeast strains and antibodies. This work is dedicated to the memory of Dr. Cohue Peña, who unexpectedly passed away. We thank all members of the Panse laboratory for enthusiastic discussions. We thank the services of the Center for Microscopy and Image Analysis, University of Zurich for providing and maintaining the imaging equipment.

## Additional information

### Funding

| Funder | Grant reference number | Author |
|---|---|---|
| Schweizerischer Nationalfonds zur Förderung der Wissenschaftlichen Forschung | | Vikram G Panse |
| H2020 European Research Council | EURIBIO | Vikram G Panse |
| Novartis StiftungfürMedizinisch-Biologische Forschung | | Vikram G Panse |

The funders had no role in study design, data collection and interpretation, or the decision to submit the work for publication.

### Author contributions

Purnima Klingauf-Nerurkar, Olga T Schubert, Yiming Chang, Conceptualization, Formal analysis, Investigation, Methodology; Ludovic C Gillet, Conceptualization, Data curation, Formal analysis, Methodology; Daniela Portugal-Calisto, Agnese Pisano, Formal analysis; Michaela Oborská-Oplová, Sanjana Rao, Formal analysis, Methodology; Martin Jäger, Formal analysis, Investigation, Methodology; Cohue Peña, Martin Altvater, Conceptualization, Formal analysis, Methodology; Ruedi Aebersold, Conceptualization, Resources, Supervision, Methodology, Writing - review and editing; Vikram G Panse, Conceptualization, Formal analysis, Supervision, Investigation, Project administration

### Author ORCIDs

Ludovic C Gillet http://orcid.org/0000-0002-1001-3265
Daniela Portugal-Calisto http://orcid.org/0000-0003-1591-5812
Michaela Oborská-Oplová https://orcid.org/0000-0003-0976-4341
Olga T Schubert https://orcid.org/0000-0002-2613-0714
Vikram G Panse https://orcid.org/0000-0001-7950-5746

### Decision letter and Author response

Decision letter https://doi.org/10.7554/eLife.52474.sa1
Author response https://doi.org/10.7554/eLife.52474.sa2

## Additional files

### Supplementary files

• Supplementary file 1. List of yeast strains used in this study.

• Supplementary file 2. List of plasmids used in this study.

• Supplementary file 3. Quantification of microscopy data. The average fraction of >1000 cells showing stronger nuclear fluorescence than cytoplasmic ($f_n$) from three independent experiments.

• Transparent reporting form

### Data availability

The mass spectrometry data reported in this study has been deposited into the ProteomeXchange Consortium via the PRIDE partner repository with dataset identifier PXD011382.

The following dataset was generated:

| Author(s) | Year | Dataset title | Dataset URL | Database and Identifier |
|---|---|---|---|---|
| Purnima Klingauf-Nerurkar, Ludovic C Gillet, Daniela Portugal-Calisto, Michaela Oborská-Oplová, Martin Jäger, Olga T Schubert, Agnese Pisano, Cohue Peña, Sanjana Rao, Martin Altvater, Yiming Chang, Ruedi Aebersold, Vikram G Panse | 2018 | The GTPase Nog1 co-ordinates assembly, maturation and quality control of distant ribosomal functional centers | http://proteomecentral. proteomexchange.org/ cgi/GetDataset?ID= PXD011382 | ProteomeXchange, PXD011382 |

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
