## [Decision Letter]

**Acceptance summary:**

The production of ribosomes relies on a fundamental and intricate assembly platform that is dependent on numerous assembly factors. This manuscript finely details the role of the GTPase Nog1 in late cytoplasmic stages of maturation of the large ribosomal subunit. The authors utilized SWATH-MS to delineate the temporal association of assembly factors with pre-60S particles. The mass spectrometry results compliment recent cryo-EM structures of pre-60S particles but also reveal insight into several assembly factors not yet visualized on pre-60S particles. Through additional genetic and cell-based assays the authors conclude that Nog1 directs both pre-60S maturation and quality control in the cytoplasm. This work lays the foundation for defining the molecular mechanisms of interplay between the various energy-consuming assembly factors required for ribosome production.

**Decision letter after peer review:**

[Editors’ note: a previous version of this study was rejected after peer review, but the authors submitted for reconsideration. The first decision letter after peer review is shown below.]

Thank you for submitting your work entitled "The GTPase Nog1 couples polypeptide exit tunnel quality control with ribosomal stalk assembly" for consideration by *eLife*. Your article has been reviewed by three peer reviewers, one of whom is a member of our Board of Reviewing Editors, and the evaluation has been overseen by a Senior Editor. The following individual involved in review of your submission has agreed to reveal their identity: Arlen Johnson (Reviewer #2).

Our decision has been reached after consultation between the reviewers. Based on these discussions and the individual reviews below, we regret to inform you that your work will not be considered further for publication in *eLife*.

While there was much in the work that all three reviewers found commendable, two of them were adamant that it was imperative that additional experiments be conducted to truncate the Nog1-CTT and determine whether the outcomes support the predictions of your coupling model, of preventing Rei1 probing of PET integrity and blocking Arx1 release. However, because the Nog1-CTT is dispensable, there is a strong possibility that the CTT truncations would not have the predicted effects, in which event, a major conclusion of the work (described in the Abstract) would be unsupported. It was also felt that you should examine whether or not the MRT4-G68E mutant allows stalk assembly, and also probe the importance of the Rlp24-Nog1 interface; although neither of these deficiencies was judged to be as serious as the lack of information about the consequences of truncating the Nog1-CTT. In the event you choose to perform additional requested experiments, and also revise the paper extensively to address other criticisms, an effort would be made to have the revised manuscript, representing a new submission, reviewed by the same three reviewers who examined the current version of the work.

*Reviewer:*

This paper uses a combination of mass spectrometry (MS), genetics, molecular and cell biology to provide evidence regarding the biochemical functions of the ATPase Drg1 and GTPase Nog1 in late steps of 60S ribosome biogenesis. They employ dominant-negative (DN) alleles of the respective genes and examine the consequences of inactivating each protein on the association of biogenesis factors with different affinity-purified pre-60S complexes, or nucleolar vs. cytoplasmic localization of shuttling biogenesis factors. In this way they show that the NOG1-DN product, altered in the G-domain to impair GTP hydrolysis, accumulates on late pre-60Ss found in the cytoplasm harboring Lsg1 and is depleted from early nucleolar complexes containing Ssf1. They present results using SWATH-S MS analysis providing quantitative estimates of the occupancies of various biogenesis factors in the Ssf1, Lsg1, and also Rix1 and Arx1 containing complexes, purified using TAP-tagged forms of each protein. Applying the same analysis to the Lsg1 complex in WT vs. DRG1-DN or NOG1-DN cells leads them to conclude that ATP hydrolysis by Drg1 is required for its own dissociation and also that of Rlp24, Bud20, Nog1, Mrt4, Nmd3 and Tif6 from the Lsg1-bound pre-60S particles, and that GTP hydrolysis by Nog1 is required for release of Mrt14, Nmd3, and Tif6 in addition to itself, but not for release of Drg1, Rlp24 and Bud20. In addition, association of Yvh1 with this complex requires the activities of both Drg1 and Nog1. These conclusions for NOG1-DN were supported by imaging analysis of GFP-tagged versions of Rpl24, Bud20, Nug1, Mrt4, Nmd3 (a variant lacking its NES) and Tif6, leading them to conclude that Nog1 GTPase is not required to initiate cytoplasmic maturation promoted by Drg1 (release of Rlp24/Bud20) but is required for subsequent steps dependent on Yvh1 of Mrt4 release and assembly of the 60S stalk and the terminal steps of Nmd3 and Tif6 release. Other results indicate that Nog1 GTPase activity is not required for maturation of the peptide exit tunnel (PET), involving insertion of the Rei1 CTT into the tunnel and release of Aix1

1) This is a difficult paper to read, partly because of the complexity of the bifurcating processing pathways, but also because the authors don't always do a good job of reminding the reader of the rationale behind experiments or what results of individual experiments indicate about the pathway. They also sometimes selectively cite results described in the figures and on a few occasions describe results in misleading ways. More care should be taken to fully explain the rationale of each experiment, describe all of the relevant results, and draw an interim conclusion before going on to the next question. It could also be helpful to cite the final model in Figure 7 in the Results section as they go along, integrating each result in the final model rather than waiting until the Discussion. There are also numerous grammatical mistakes.

2) It seems important to confirm the MS data for DRG1-DN by Western analysis of the same set of proteins analyzed in Figure 3C, plus Nog1, to support key postulates of the final model.

3) It's also unclear why the authors have not imaged the panel of GFP-tagged proteins analyzed in Figure 4 for the NOG1-DN in the strain expressing the DRG1-DN allele, to confirm that localization of Bud20, Nug1, and Arx1 would be altered in addition to that of Mrt4, Nmd3 and Tif6 by inactivating Drg1. Some of this is accomplished later with the orthogonal approach of inhibiting Drg1 with diazaborine, and perhaps some of it was published previously, and if so, should be cited at the appropriate place in the Results.

4) There is a noteworthy discrepancy between the MS and Western data for Arx1 that should be discussed/resolved. Shouldn't both DN alleles lead to accumulation of Arx1 in the Lsg1-bound intermediate, whereas none is evident in the MS data of Figure 3B?

*Reviewer #2:*

This manuscript proposes that the GTPase Nog1, involved in ribosome assembly, couples quality control of the polypeptide exit tunnel (PET) with assembly of the P0 stalk. Nog1 is an unusual protein in that its N-terminus disrupts the ribosomal A site while its C-terminus wraps around the pre-60S particle and inserts into the PET. The authors use a dominant negative mutation in Nog1 that likely blocks GTP hydrolysis, although this is not shown. The major conclusion of this work is that the dominant negative mutant cannot be released from the ribosome and blocks the recruitment of Yvh1, preventing assembly of the P0 stalk but does not block events at the polypeptide exit tunnel. The data are consistent with the model in which the AAA-ATPase Drg1 initiates cytoplasmic maturation of the pre-60S by removing Rlp24, and probably the CTT of Nog1 at the same time. This is followed by release of Nog1 from the A site, providing linkage between events at the PET and P0 stalk. However, such coupling is not rigorously established. In addition, the function of Nog1 "probing" the exit tunnel seems dispensable because GFP fusion and C-terminal truncations of Nog1, which would preclude probing, are functional. This leaves open the question of the functional significance of Nog1 insertion into the PET. Lastly, the insertion of Nog1 into the A site disrupts the PTC, the catalytic center of the ribosome. The authors should also suggest that Nog1 couples probing of the PET with completion of the catalytic center.

1) The authors propose that Nog1 couples events probing the PET with stalk assembly. Additional experiments should be done to better establish this coupling. The C-terminus of Nog1 is dispensable. The authors should test if nog1∆CTT uncouples these events. Does nog1∆CTT allow stalk assembly in the presence of DRG1-DN?

2) "pre-60S particle requires successful completion of PET maturation/quality control and ribosomal stalk assembly" True but it also requires completion of the PTC! That's the all-important catalytic center.

3) Subsection “Nog1^DN^ blocks the terminal cytoplasmic maturation steps”. The authors show that Mrt4^G68E^ localizes properly in the presence of Nog1^DN^ but does not rescue the inviability. But does it allow assembly of the P0 stalk?

*Reviewer #3:*

The manuscript by Klingauf-Merurkar et al. nicely clarifies the details of several steps of cytoplasmic maturation of the large ribosomal subunit. Ribosome assembly is a very complicated pathway that is dependent upon numerous energy consuming enzymes such as the AAA-ATPase Drg1 and the GTPase Nog1. The essential GTPase Nog1 binds to the pre-60S in a very unusual way with its C-terminus inserting into the PET. Nog1 also intertwines with Rlp24, the substrate for Drg1. To study the roles of Nog1 and Drg1 Klingauf-Merurkar et al. used dominant negative mutants of both enzymes, previously established by others. The authors isolated well characterized pre-60S particles by TAP purification methods and carried out SWATH-S mass spectrometry. This mass spectrometry approach agrees with earlier studies but allows for quantification and the identification of ribosome assembly factors not characterized in recent cryo-EM structures. SWATH-S mass spectrometry of pre-60S particles isolated with the Drg1 and Nog1 dominant negative mutants led to the observation that the Nog1 mutant did not accumulate Rlp24 and Bud20, which both accumulate with the Drg1 mutant. The authors confirm their mass spectrometry results with TAP-purifications coupled to western blots and localization studies. They conclude that the activities of Nog1 and Drg1 are stepwise. ATP hydrolysis by Drg1 releases Rlp24 and Bud20 from pre-60S particles, as well as extracts the Nog1 CTT from the PET. Next GTP hydrolysis by Nog1 stimulates its removal from the pre-60S allowing later acting assembly factors to bind.

1) The authors conclude that ATP hydrolysis by Drg1 releases the Nog1 CTT from the PET but they provide no direct evidence for this. This is inferred from accumulation of later stage assembly factors only with the Nog1-DM mutant. The authors should test this hypothesis by carrying out experiments with a series of Nog1- CTT truncations.

2) The authors suggest that Rlp24 might serve as an adapter for Drg1 to extract the Nog1 CTT from the PET but there is no evidence to suggest that Nog1 is a direct substrate of Drg1. Previous studies have shown that the C-terminus of Rlp24 stimulates Drg1 ATP hydrolysis, strongly suggesting that Rlp24 is the substrate for remodeling by Drg1 not Nog1. Based on the cryo-EM structure by Wu et al. it seems a more likely scenario that remodeling/unfolding of Rlp24 by Drg1 leads to the removal of the Nog1 CTT from the PET. The authors should further address this by probing the Rlp24-Nog1 interface to disrupt the known binding interfaces in both proteins.

3) There is no mention in the manuscript of Ribosomal Protein L24. Rlp24 acts as a placeholder for L24 on pre-60S particles. If Rpl24 is released by Drg1 prior to release of Nog1, it seems possible that L24 could associate with Nog1 prior to its release. Do the authors detect L24 in their mass spectrometry analysis?

4) A novel finding in the manuscript is the identification of previously uncharacterized assembly factors such as Tma16 that accumulate with the Drg1 and Nog1 mutants. The authors should further explore the function of Tma16 and its links to the activities of Drg1 and Nog1.

[Editors’ note: what now follows is the decision letter after the authors submitted for further consideration.]

Thank you for submitting your article "The GTPase Nog1 co-ordinates assembly, maturation and quality control of distant ribosomal functional centers" for consideration by *eLife*. Your article has been reviewed by three peer reviewers, one of whom is a member of our Board of Reviewing Editors, and the evaluation has been overseen by James Manley as the Senior Editor. The following individuals involved in review of your submission have agreed to reveal their identity: Arlen Johnson (Reviewer #2); Alan John Warren (Reviewer #3).

The reviewers have discussed the reviews with one another and the Reviewing Editor has drafted this decision to help you prepare a revised submission.

Summary:

This is an interesting study that combines SWATH-mass spectrometry with genetic and cell biological assays to study the role of the ATPase Drg1 and the GTPase Nog1 in late-stage cytoplasmic steps of 60S ribosomal subunit maturation. All three reviewers commended the authors for addressing the major concerns of the initial submission with the inclusion of key additional experiments. However, all three reviewers feel that revisions are needed to the figures and text to make the manuscript accessible to a general audience. The text requires a number of updates/corrections to take account of recently published work and citations need to be added appropriately within the text. Reviewers also raised concerns about the interpretation of some of the data. Based on our joint discussion we invite you to submit a suitably revised version of your manuscript in which you address the points raised below.

Essential revisions:

1) Data/Results Interpretation:

1a) This manuscript is missing any sort of quantification of the immunofluorescence data, which is present in 5 figures. According to the Materials and methods section each experiment was performed on three different occasions and in triplicate with a sample size of >1000 cells. The authors should quantify the number of cells with nuclear or cytoplasmic accumulation and present this data in a supplementary table.

1b) In Figure 3 the authors compare the SWATH-MS results with western blot analysis only for Nog1-DN. Antibodies may not be available for all the proteins shown by mass spectrometry but it's unclear why the authors do not show the mass spectrometry data for the ribosomal proteins that they analyze by western blot (uL10, eL24, and uL29). Comparing panels B and C would also be much easier if the western blots were shown in the same order as the mass spectrometry.

1c) Did the authors identify eL40 in the SWATH-MS data as this ribosomal protein is clearly critical for late cytoplasmic 60S maturation (Fernández-Pevida et al., 2012). eL40 should also be included in the "revised model for cytoplasmic maturation of the 60S ribosomal subunit".

1d) Subsection “Nog1^DN^ blocks the terminal cytoplasmic maturation steps”: the authors reference Weis et al., 2015 to support "tightly coupled co-release of Nmd3 and Tif6". It would be more accurate to reference work from the Johnson lab in this regard (Lo et al., 2011, Malyutin et al., 2017; Patchett et al., 2017). We suggest that current cryo-EM data instead support the sequential removal of Nmd3 and Tif6 (Weis et al., 2015; Ma et al., 2017; Kargas et al., 2019). Please amend the text (and Figure 8, which is ambiguous in terms of Nmd3/Tif6 removal).

1e) Subsection “Nog1^DN^ blocks the terminal cytoplasmic maturation steps”: "co-release of Nmd3 and Tif6…" This statement is not consistent with recent cryo-EM data (Kargas et al) as discussed above. Please update/discuss.

1f) Subsection “Nog1^DN^ blocks the terminal cytoplasmic maturation steps”: the authors reference Weis et al., 2015 to support "tightly coupled co-release of Nmd3 and Tif6". It would be more accurate to reference work from the Johnson lab in this regard (Lo et al., 2011, Malyutin et al., 2017; Patchett et al., 2017). We suggest that current cryo-EM data instead support the sequential removal of Nmd3 and Tif6 (Weis et al., 2015; Ma et al., 2017; Kargas et al., 2019). Please amend the text (and Figure 8, which is ambiguous in terms of Nmd3/Tif6 removal).

1g) "Given the tightly coupled release of Nmd3 and Tif6, as expected…". Why is it necessary to invoke "tight coupling" here?

1h) Subsection “Nog1^DN^ blocks the terminal cytoplasmic maturation steps”: A key problem caused by the retention of Nog1^DN^ is that it prevents rearrangement of H89, providing a steric barrier to the recruitment of ribosomal protein uL16. The uL16 protein reorientates rRNA helices H89 and H38 to promote conformational rearrangement of the Nmd3 C-terminus, not the N-terminus (Kargas et al., 2019). The Nmd3 N-terminus remains fixed to Tif6 throughout the late steps of maturation while the C-terminal domain undergoes re-arrangement following the reorientation of H38 (Kargas et al., 2019). Please update this section.

2) The Discussion is long and somewhat repetitive with the Results. Please add appropriate subheadings to make this easier to follow. The Discussion must address the recent findings highlighted below and the proposed "revised model" should be clarified either to include all the other known cytoplasmic assembly factors or to explain clearly why specific factors have been excluded.

2a) In Figure 1C, the Western blot indicates that wild type Nog1 binds to the Lsg1-TAP particle, at least when Nog1 is over-expressed from the GAL promoter. Figure 2 also suggests that at least a subset of Lsg1-TAP particles bind Nog1, while the genetic experiments in Figure 3B clearly show that Nog1 accumulates on the Lsg1-TAP particle either when Drg1 function is impaired or in the presence of the Nog1^DN^ mutation. These data suggest that Nog1 and Lsg1 likely bind simultaneously to a subset of native cytoplasmic pre-60S particles. This point should be discussed and if appropriate, Figure 8 updated.

2b) Using cryo-EM, Zhou et al., 2019, identify an early cytoplasmic pre-60S particle in which Bud20 is bound, the Nog1 CTT is in the PET, but the N-terminal and G-domains of Nog1 have apparently been displaced from the ribosome. Please discuss in the text how the SWATH-MS and genetic data presented in the current manuscript are consistent with the cryo-EM maps by Zhou et al?

2c) The SWATH-MS data in Figures 2 and 3 are consistent with recent cryo-EM data (Ma et al., 2017; Kargas et al., 2019) that reveal sequential recruitment of Rei1 and Reh1 into the PET. However, Reh1 is not included in the "revised model" in Figure 8. Please discuss/amend.

2d) In Figure 3B, the left hand panel- consistent with the cryo-EM structures from Kargas et al., 2019, another possible interpretation of the SWATH-MS data might be that like Tif6 and Nmd3, Reh1 is simply not recycled from late pre-60S particles in the Nog1^DN^ or Drg1-deficient cells. It therefore seems unnecessary to invoke the recruitment of Reh1 and Rei1 through distinct binding sites. Please discuss. Perhaps it would be more accurate for the authors to indicate explicitly that they are proposing a revised model for the earlier steps of cytoplasmic pre-60S maturation.

2e) Recent work (Su et al., 2019) has identified the structure and function of Vms1 and Arb1 in RQC and mitochondrial proteome homeostasis. Please comment how the findings on Arb1 distribution in pre-60S particles in the current manuscript might relate to the identified function of Arb1 in RQC and update the text accordingly. It is inaccurate to state "structural information in the context of the ribosome is lacking".

2f) The Discussion should be updated to take account of recent cryo-EM data (Kargas et al., 2019) which reveals that in fact the N-terminus of Nmd3 remains fixed to Tif6 during the later steps of 60S maturation-it is the C-terminal domain that undergoes conformational rearrangement upon uL16 binding and H38 rearrangement.

2g) Figure 8A: Given that this is a "revised model for cytoplasmic maturation of the 60S ribosomal subunit", Lsg1, Reh1 and eL40 should also be included. In the figure legend, the statement "Nog1 triggers its own release together with Nsa2" does not appear to have been tested experimentally in the paper. Also, "we suggest that co-release of Nog1 and Nsa2…" Please provide evidence to support these statements or modify the text.

2h) Figure 8B: The typos in the legend should be corrected (H89, not H89) and the meaning of the text clarified. What rearrangement of Nmd3 is being referred to here? In the central panel, why is Rei1-CTT and not Reh1-CTT indicated? uL16 is recruited after the exchange of Rei1 for Reh1 (Kargas et al., 2019). Is this figure really necessary? It does not seem to directly relate to the work described in the manuscript and is not particularly informative.

---

## [Author Response]

[Editors’ note: the author responses to the first round of peer review follow.]

Reviewer #1:[…] 1) This is a difficult paper to read, partly because of the complexity of the bifurcating processing pathways, but also because the authors don't always do a good job of reminding the reader of the rationale behind experiments or what results of individual experiments indicate about the pathway. They also sometimes selectively cite results described in the figures and on a few occasions describe results in misleading ways. More care should be taken to fully explain the rationale of each experiment, describe all of the relevant results, and draw an interim conclusion before going on to the next question. It could also be helpful to cite the final model in Figure 7 in the Results section as they go along, integrating each result in the final model rather than waiting until the Discussion. There are also numerous grammatical mistakes.

We thank the reviewer for his/her constructive comments. In the revised version of the manuscript, we now better explain how different domains of the GTPase Nog1 regulate PET maturation, PTC formation and stalk assembly.

2) It seems important to confirm the MS data for DRG1-DN by Western analysis of the same set of proteins analyzed in Figure 3C, plus Nog1, to support key postulates of the final model.

Western analyses validating accumulation of specific assembly factors, including Nog1, on the Lsg1-TAP particle in DRG1-DN expressing cells has been reported by several labs (Pertschy et al., 2007; Lo et al., 2010), including ours (Altvater et al., 2012).

3) It's also unclear why the authors have not imaged the panel of GFP-tagged proteins analyzed in Figure 4 for the NOG1-DN in the strain expressing the DRG1-DN allele, to confirm that localization of Bud20, Nug1, and Arx1 would be altered in addition to that of Mrt4, Nmd3 and Tif6 by inactivating Drg1. Some of this is accomplished later with the orthogonal approach of inhibiting Drg1 with diazaborine, and perhaps some of it was published previously, and if so, should be cited at the appropriate place in the Results.

Several laboratories, including ours, have reported mislocalization of Bud20, Nug1 and Arx1 in DRG1-DN expressing cells (or diazaborine treatment) into the cytoplasm (Pertschy et al., 2007; Lo et al., 2010; Altvater et al., Loibl et al., 2014). We cite the relevant references at appropriate places.

Here, we have localized Bud20, Nug1 and Arx1 in NOG1-DN expressing cells treated with diazaborine (Figure 5D). As expected, Bud20, Nug1 and Arx1 mislocalize into the cytoplasm only after treating NOG1-DN expressing cells with diazaborine.

4) There is a noteworthy discrepancy between the MS and Western data for Arx1 that should be discussed/resolved. Shouldn't both DN alleles lead to accumulation of Arx1 in the Lsg1-bound intermediate, whereas none is evident in the MS data of Figure 3B?

Arx1 bound to the pre-60S, mislocalises to the cytoplasm in DRG1-DN expressing cells (Altvater et al., 2012.; Lo et al., 2010). This is because recruitment of the Arx1-release factor, Rei1, to the pre-60S in the cytoplasm is impaired in DRG1-DN expressing cells (Altvater et al., 2012; Lo et al., 2010). Cryo-EM studies show that the Rei1-binding site on the pre-60S clashes with the Rlp24:Nog1-CTT complex (Figure 5C), providing an explanation as to why Drg1-mediated Rlp24 release is critical to recruit Rei1 for PET maturation (Arx1 release).

In contrast, recruitment of Rei1 to the pre-60S is unaffected in NOG1-DN expressing cells (Figure 3B, 3C and Figure 5A). Moreover, Arx1 release from the pre-60S is also not impaired (Figure 4C). This means that Rei1-CTT has successfully probed the PET in these cells. Failure to insert the Rei1-CTT into PET by attaching a TAP tag blocks Arx1 release, and mislocalizes Arx1 to the cytoplasm (Greber et al., 2016; Figure 5D). This mislocalisation is also observed in the rei1-TAP cells expressing NOG1-DN (Figure 5D) supporting the idea that Rei1 probes PET integrity in NOG1-DN expressing cells, and that Nog1-DN-CTT has to be removed from the PET.

Arx1 MS discrepancy: We have observed a similar discrepancy in Arx1 quantification in previous SRM-MS analyses (Altvater et al., 2012). The reasons for this discrepancy remain unclear. To resolve this, we have employed Arx1-GFP location as a reporter for PET maturation.

Reviewer #2:This manuscript proposes that the GTPase Nog1, involved in ribosome assembly, couples quality control of the polypeptide exit tunnel (PET) with assembly of the P0 stalk. Nog1 is an unusual protein in that its N-terminus disrupts the ribosomal A site while its C-terminus wraps around the pre-60S particle and inserts into the PET. The authors use a dominant negative mutation in Nog1 that likely blocks GTP hydrolysis, although this is not shown. The major conclusion of this work is that the dominant negative mutant cannot be released from the ribosome and blocks the recruitment of Yvh1, preventing assembly of the P0 stalk but does not block events at the polypeptide exit tunnel. The data are consistent with the model in which the AAA-ATPase Drg1 initiates cytoplasmic maturation of the pre-60S by removing Rlp24, and probably the CTT of Nog1 at the same time. This is followed by release of Nog1 from the A site, providing linkage between events at the PET and P0 stalk. However, such coupling is not rigorously established. In addition, the function of Nog1 "probing" the exit tunnel seems dispensable because GFP fusion and C-terminal truncations of Nog1, which would preclude probing, are functional. This leaves open the question of the functional significance of Nog1 insertion into the PET. Lastly, the insertion of Nog1 into the A site disrupts the PTC, the catalytic center of the ribosome. The authors should also suggest that Nog1 couples probing of the PET with completion of the catalytic center.1) The authors propose that Nog1 couples events probing the PET with stalk assembly. Additional experiments should be done to better establish this coupling. The C-terminus of Nog1 is dispensable. The authors should test if nog1∆CTT uncouples these events. Does nog1∆CTT allow stalk assembly in the presence of DRG1-DN?

In addition to the N-terminal domain (NTD) and a G-domain, Nog1 contains a long Cterminal tail (CTT). The Nog1-CTT intertwines around the ribosomal like protein Rlp24, then makes contacts with Arx1, and finally inserts its terminal end into the PET (Figure 1A). We have analysed the phenotypes of a series of Nog1-CTT truncation mutants. Removal of the part of the CTT (Nog1^1-536^ and Nog1^1-479^) that is inserted into the PET is not lethal, but renders a cold sensitive phenotype (Figure 6B). Likewise, attachment of a large tag GFP tag to Nog1CTT is not lethal, but renders a growth defect, stronger than the CTT truncations (Figure 6B). Sucrose density gradient analyses of lysates derived from the Nog1-GFP strain show the presence of halfmers that characterise 40S and 60S subunit imbalance (Author response image 1). Removal of the entire Nog1-CTT that includes the region that intertwines with Rlp24 (Nog1^1-426^) is lethal in yeast (Figure 6B).

**Author response image 1. respfig1:** Functional analyses of yeast mutants expressing Nog1-GFP and Nog1-FLAG. (**A**) Polysome profile analysis of Nog1-GFP. Wild-type and Nog1-GFP strains were grown in YPD at 30°C to mid-log-phase. Cell extracts were prepared after cycloheximide treatment and subjected to sedimentation centrifugation on 7-50% sucrose density gradients (Altvater et al., 2012). Halfmers are indicated with asterisks. (**B**) Left panel: Work flow depicting sequential ProteinA (Lsg1-TAP) and FLAG purification (Nog1^DN^-FLAG) from the indicated strains. Middle panel: Nog1^DN^-FLAG tail remains accessible for purification. Lsg1-TAP strain was transformed with either vector or Nog1^DN^FLAG under GAL1 promoter. Subsequently, either vector or with Drg1^DN^ under GAL1 promoter were transformed. The resultant strains were grown in raffinose-containing synthetic medium and induced with 2% galactose for 3 hours. After sequential ProteinA and FLAG purifications, the eluates were analyzed by Western blotting using indicated antibodies. Right panel: Nog1-FLAG strain is growth impaired at 20 and 25 degrees. A nog1Δ shuffle strain was transformed with plasmids encoding either Nog1 or Nog1-FLAG plasmids. After shuffling out the URA3-NOG1 plasmid on FOA-containing plates, the resultant strains were spotted in 10-fold serial dilutions on YPD plates and incubated for 2-4 days at the indicated temperatures.

We apologize to the reviewer for not clearly explaining how different domains of Nog1 coordinate PET maturation and stalk assembly. PET maturation is impaired in DRG1-DN expressing cells. This is because the Rei1-binding site on the pre-60S clashes with the Rlp24:Nog1-CTT complex (Figure 5C). Drg1-driven Rlp24 release from the pre-60S (consequently Nog1-CTT) is critical for Rei1 recruitment (Lo et al., 2010; Altvater et al., 2012). In addition, Rei1 also needs to insert its CTT end into the PET to probe its integrity, to permit Arx1 release. For this to occur, Nog1-CTT has to be removed from the PET. Given that an impaired Nog1-G domain does not interfere with PET maturation implies that Rei1 has inserted its CTT into the PET. NOG1-DN expression in *rei1-TAP* strain impairs PET maturation supporting the notion that Rei1-CTT probes the PET in these cells, again implying Nog1-CTT is out (Figure 5D).

As requested, we investigated PET maturation (Arx1-GFP) after expressing DRG1-DN in a Nog1-CTT truncation mutant, Nog1^1-479^, that still retains its Rlp24 binding site (Figure 6D). In light of cryo-EM data (Figure 5C), as expected, Arx1-GFP mislocalises to the cytoplasm in the Nog1-truncation mutant expressing DRG1-DN (Figure 6D). The Yvh1 binding site on the pre-60S clashes with the Nog1-G domain, providing an explanation as to why NOG1-DN expressing cells are impaired in stalk assembly (Figure 4D).

It is the negative regulation by the Nog1-CTT:Rlp24 complex in PET maturation (Rei1 recruitment) and stalk assembly (Yvh1 recruitment) that led to the proposal that Nog1 couples assembly and quality control of distant ribosomal functional centres.

2) "pre-60S particle requires successful completion of PET maturation/quality control and ribosomal stalk assembly" True but it also requires completion of the PTC! That's the all-important catalytic center.

We thank the reviewer for pointing this out. We now stated this in the Abstract and incorporated this important point in the Discussion.

3) Subsection “Nog1^DN^ blocks the terminal cytoplasmic maturation steps”. The authors show that Mrt4^G68E^ localizes properly in the presence of Nog1^DN^ but does not rescue the inviability. But does it allow assembly of the P0 stalk?

We have performed Western analyses to investigate this point. The ribosomal stalk protein P0 (uL10) is recruited to the 60S pre-ribosome upon expression of Nog1-DN in the MRT4^G68E^ strain as judged by Western analyses (Figure 7C). Yes, it does allow assembly of the P0 stalk.

Reviewer #3:[…] 1) The authors conclude that ATP hydrolysis by Drg1 releases the Nog1 CTT from the PET but they provide no direct evidence for this. This is inferred from accumulation of later stage assembly factors only with the Nog1-DM mutant. The authors should test this hypothesis by carrying out experiments with a series of Nog1- CTT truncations.

We apologize to the reviewer for not clearly explaining how Nog1 regulates PET maturation and stalk assembly.

In addition to the N-terminal domain (NTD) and a G-domain, Nog1 exhibits an extensive C-terminal tail (CTT). The CTT: (1) intertwines around the ribosomal like protein Rlp24 (2) contacts the assembly factor Arx1, before (3) inserting its terminal end into the PET (Figure 1A). We constructed Nog1-CTT truncation mutants analysed their phenotypes in yeast (Figure 6B). Removal of the part of the CTT that is inserted into the PET (Nog1^1-536^ and Nog1^1479^) is not lethal, but renders a cold sensitive phenotype (Figure 6B). Likewise, attachment of a large tag GFP tag is not lethal, but renders a growth defect even stronger than the Nog1CTT truncation mutants (Figure 6B). Further truncation of the Nog1-CTT that includes the region that intertwines with Rlp24 (Nog1^1-426^) is lethal (Figure 6A, 6B).

PET maturation is impaired in DRG1-DN expressing cells (Pertschy et al., 2007; Lo et al., 2010; Altvater et al., 2012). This is because Rei1 recruitment to the pre-60S is impaired in these cells (Lo et al., 2010; Altvater et al., 2012). Cryo-EM studies indicate that the Rei1 binding site on the pre-60S clashes with the Rlp24:Nog1-CTT complex (Figure 5C) providing an explanation as to why Drg1-driven Rlp24 release from the pre-60S (and consequently Nog1-CTT) is critical for Rei1 recruitment.

It is also critical for Rei1 to insert its CTT into the PET, to permit Arx1 release and complete PET maturation (Greber et al., 2016). This can only occur if Nog1-CTT has been extracted from the PET. Based on two different observations that: (1) an impaired Nog1G domain does not interfere with PET maturation and (2) NOG1-DN expression only in the rei1-TAP mutant (where Rei1 is unable to insert its tail into tunnel) impairs PET maturation (Figure 5D), we propose that Rei1 has inserted its CTT into the PET, and therefore infer that NOG1-DN-CTT has been removed from the PET.

We investigated PET maturation (Arx1-GFP) in the Nog1^1-479^ truncation mutant expressing DRG1-DN. The Nog1^1-479^ truncation mutant retains the region that intertwines with Rlp24. In agreement with cryo-EM studies (Figure 5C), Arx1-GFP mislocalises to the cytoplasm in Nog1^1-479^ cells treated with DIA (Figure 6D) suggesting that removal of Rlp24 from the pre-60S (and consequently Nog1-CTT) is critical for Rei1 recruitment.

The Yvh1 binding site clashes with the Nog1-G domain, thus providing structural basis as to why NOG1-DN expressing cells are impaired in stalk assembly (Figure 4D). It is the negative regulation of PET maturation and stalk assembly through the Nog1-CTT:Rlp24 complex and Nog1-G domain, respectively, that led to the proposal that Nog1 couples PET maturation and stalk assembly.

Our inference that Drg1-ATPase activity releases Nog1-CTT terminal end from the PET is based on a combination of biochemical and cell-biological observations: DRG1-DN blocks PET maturation by not recruiting the Arx1-release factor Rei1. Although NOG1-DN remains on the cytoplasmic pre-60S, it does not block Rei1 recruitment, as well as insertion of Rei1-CTT into the PET. Consequently, Arx1 release and recycling is not impaired. If NOG1^DN^-CTT had remained inserted inside the PET, then it would have blocked Arx1-release. As the reviewer correctly points out the likely explanation is that: Drg1 mediated removal of Rlp24 from the Nog1-CTT:Rlp24 complex leads to the removal of Nog1-CTT from the PET, thus permitting Rei1 recruitment and PET probing for tunnel maturation.

By employing tandem Lsg1-TAP followed by Nog1^DN^-CTT-FLAG purifications (work flow in Author response image 1). we have attempted to directly demonstrate that the Nog1^DN^-CTT-FLAG is not inserted into the tunnel. Unfortunately, Nog1-FLAG fusion construct is not fully functional and renders a cold sensitive phenotype to yeast in a manner very similar to the one seen for the viable Nog1-CTT truncations, suggesting that the FLAGtagged tail might not be inserted into the PET. Consistent with this, we found that Nog1^DN^CTT-FLAG tail is accessible for affinity purification, that is it is not inserted in the PET upon Drg1^DN^ expression (Lane 3). Note that the assembly factor Bud20 that interacts with Rlp24 coenriches with this particle, showing that Drg1^DN^ expression blocked initiation of the cytoplasmic maturation pathway in these cells.

2) The authors suggest that Rlp24 might serve as an adapter for Drg1 to extract the Nog1 CTT from the PET but there is no evidence to suggest that Nog1 is a direct substrate of Drg1. Previous studies have shown that the C-terminus of Rlp24 stimulates Drg1 ATP hydrolysis, strongly suggesting that Rlp24 is the substrate for remodeling by Drg1 not Nog1. Based on the cryo-EM structure by Wu et al. it seems a more likely scenario that remodeling/unfolding of Rlp24 by Drg1 leads to the removal of the Nog1 CTT from the PET. The authors should further address this by probing the Rlp24-Nog1 interface to disrupt the known binding interfaces in both proteins.

Nog1 is not a direct substrate of Drg1; if this were the case it would have been completely removed from the pre-60S, without the need for the GTPase activity. Rlp24 directly binds to Drg1, stimulates its ATPase activity for release from the pre-60S. Given that parts of the Nog1-CTT intertwine around Rlp24 on the pre-60S, the reviewer is correct to point out that the extraction of Rlp24 by Drg1 is likely to lead Nog1-CTT removal from the PET.

The Drg1 “adaptor” suggestion put forth in the Discussion, stems from the model AAAATPase, Cdc48/p97. This AAA-ATPase employs different adaptor proteins to remodel different protein complexes and impact diverse cellular processes. Although only Rlp24 is a direct substrate of Drg1, we proposed that it serves as an adaptor through interactions with Nog1-CTT to indirectly remove the CTT-tail end from the PET.

Removal of the Nog1-CTT region that binds the Rlp24 (Nog1^1-426^ mutant) is lethal in yeast. Overexpression of a Nog1 construct lacking the Rlp24 binding region, Nog1^∆427-536^, is dominant negative, but unlike NOG1-DN, it does not impair the 60S cytoplasmic maturation pathway (Figure 6B, 6C). Given that Nog1 and Rlp24 rely on each for their recruitment to the pre-60S (Saveanu et al., 2003), the toxicity of the Nog1^∆427-536^ very likely reflects a failure to form a functional Nog1-CTT:Rlp24 complex on the pre-60S during early nucleolar/nuclear maturation.

3) There is no mention in the manuscript of Ribosomal Protein L24. Rlp24 acts as a placeholder for L24 on pre-60S particles. If Rpl24 is released by Drg1 prior to release of Nog1, it seems possible that L24 could associate with Nog1 prior to its release. Do the authors detect L24 in their mass spectrometry analysis?

The release and recycling of Rlp24 from the 60S pre-ribosome is not impaired in NOG1-DN expressing cells. As requested, we have analysed recruitment of eL24 to the pre60S in NOG1-DN expressing cells by Western blotting. eL24 is recruited to the pre-60S in these cells (Figure 3C).

4) A novel finding in the manuscript is the identification of previously uncharacterized assembly factors such as Tma16 that accumulate with the Drg1 and Nog1 mutants. The authors should further explore the function of Tma16 and its links to the activities of Drg1 and Nog1.

By employing Western analyses, we confirmed the SWATH-MS data that Tma16 accumulates on cytoplasmic 60S pre-ribosomes in NOG1-DN expressing cells (Figure 3C). To investigate the precise location of Tma16 on the pre-60S and its functional contribution of to the cytoplasmic maturation pathway is an important issue. In our opinion, Tma16 analyses go beyond the main message of this study that sequential Drg1-ATPase and Nog1-GTPase activities license PET maturation, PTC formation and ribosomal stalk assembly.

[Editors' note: the author responses to the re-review follow.]

Summary:This is an interesting study that combines SWATH-mass spectrometry with genetic and cell biological assays to study the role of the ATPase Drg1 and the GTPase Nog1 in late-stage cytoplasmic steps of 60S ribosomal subunit maturation. All three reviewers commended the authors for addressing the major concerns of the initial submission with the inclusion of key additional experiments. However, all three reviewers feel that revisions are needed to the figures and text to make the manuscript accessible to a general audience. The text requires a number of updates/corrections to take account of recently published work and citations need to be added appropriately within the text. Reviewers also raised concerns about the interpretation of some of the data. Based on our joint discussion we invite you to submit a suitably revised version of your manuscript in which you address the points raised below.

We thank the reviewers for their helpful comments, and especially pointing out incorrect statements regarding the 60S cytoplasmic maturation pathway.

Essential revisions:1) Data/Results Interpretation:1a) This manuscript is missing any sort of quantification of the immunofluorescence data, which is present in 5 figures. According to the Materials and methods section each experiment was performed on three different occasions and in triplicate with a sample size of >1000 cells. The authors should quantify the number of cells with nuclear or cytoplasmic accumulation and present this data in a supplementary table.

Each cell-biological experiment was performed at least three times, and >90% of the cells show the reported phenotypes. The images were quantified as described previously in Lo et al., 2010: the average of the fraction of cells that show stronger nuclear fluorescence as compared the cytoplasm was quantified. We list the quantification for every image in Supplementary file 3.

1b) In Figure 3 the authors compare the SWATH-MS results with western blot analysis only for Nog1-DN. Antibodies may not be available for all the proteins shown by mass spectrometry but it's unclear why the authors do not show the mass spectrometry data for the ribosomal proteins that they analyze by western blot (uL10, eL24, and uL29). Comparing panels B and C would also be much easier if the western blots were shown in the same order as the mass spectrometry.1c) Did the authors identify eL40 in the SWATH-MS data as this ribosomal protein is clearly critical for late cytoplasmic 60S maturation (Fernández-Pevida et al., 2012). eL40 should also be included in the "revised model for cytoplasmic maturation of the 60S ribosomal subunit".

Nearly all ribosomal proteins are small, Lys/Arg-rich proteins. During sample preparation ribosomal proteins are broken down by trypsin into rather short peptides, as compared to larger assembly factors. In general, detecting and quantitating these tryptic peptides by mass spectrometry is challenging. Ribosomal proteins eL3, eL14 and u15 can be reliably quantified by SWATH-MS, and hence we have used them as “loading controls”. Further, we have validated the proteomic data by Western blotting and cell-biological approaches.

We have reorganized panels B and C to better compare the Western analyses and SWATH-MS data.

We were unable to reliably detect eL40 by SWATH-MS in WT Lsg1-TAP. As requested, we have included eL40 and uL16 in model for 60S maturation pathway in the text (subsection “Nog1 co-ordinates assembly, maturation and quality control of distant ribosomal centres”) and Figure 8.

1d) Subsection “Nog1^DN^ blocks the terminal cytoplasmic maturation steps”: the authors reference Weis et al., 2015 to support "tightly coupled co-release of Nmd3 and Tif6". It would be more accurate to reference work from the Johnson lab in this regard (Lo et al., 2011, Malyutin et al., 2017; Patchett et al., 2017). We suggest that current cryo-EM data instead support the sequential removal of Nmd3 and Tif6 (Weis et al., 2015; Ma et al., 2017; Kargas et al., 2019). Please amend the text (and Figure 8, which is ambiguous in terms of Nmd3/Tif6 removal).1e) Subsection “Nog1^DN^ blocks the terminal cytoplasmic maturation steps”: "co-release of Nmd3 and Tif6…" This statement is not consistent with recent cryo-EM data (Kargas et al) as discussed above. Please update/discuss.1f) Subsection “Nog1^DN^ blocks the terminal cytoplasmic maturation steps”: the authors reference Weis et al., 2015 to support "tightly coupled co-release of Nmd3 and Tif6". It would be more accurate to reference work from the Johnson lab in this regard (Lo et al., 2011, Malyutin et al., 2017; Patchett et al., 2017). We suggest that current cryo-EM data instead support the sequential removal of Nmd3 and Tif6 (Weis et al., 2015; Ma et al., 2017; Kargas et al., 2019). Please amend the text (and Figure 8, which is ambiguous in terms of Nmd3/Tif6 removal).1g) "Given the tightly coupled release of Nmd3 and Tif6, as expected…". Why is it necessary to invoke "tight coupling" here?

We thank the reviewers for pointing out our misconception regarding Nmd3 and Tif6 release from the pre-60S. Recent cryo-EM work (Kargas et al., 2019) indeed supports the model that Nmd3 and Tif6 are sequentially released. We have removed/amended incorrect statements and updated the model (Discussion) and in Figure 8.

1h) Subsection “Nog1^DN^ blocks the terminal cytoplasmic maturation steps”: A key problem caused by the retention of Nog1^DN^ is that it prevents rearrangement of H89, providing a steric barrier to the recruitment of ribosomal protein uL16. The uL16 protein reorientates rRNA helices H89 and H38 to promote conformational rearrangement of the Nmd3 C-terminus, not the N-terminus (Kargas et al., 2019). The Nmd3 N-terminus remains fixed to Tif6 throughout the late steps of maturation while the C-terminal domain undergoes re-arrangement following the reorientation of H38 (Kargas et al., 2019). Please update this section.

We thank the reviewers for pointing out our misconception regarding the order of uL16 incorporation, rearrangements in rRNA and Nmd3 after Nog1 release. We have corrected this in the text (Discussion).

2) The Discussion is long and somewhat repetitive with the Results. Please add appropriate subheadings to make this easier to follow. The Discussion must address the recent findings highlighted below and the proposed "revised model" should be clarified either to include all the other known cytoplasmic assembly factors or to explain clearly why specific factors have been excluded.

We have removed the repetitive sections, and shortened the Discussion. We now provide subheadings that highlight our key findings. The Discussion is focused on how different domains of Nog1 gate PET maturation/quality control, PTC maturation, and late cytoplasmic maturation steps (Discussion).

2a) In Figure 1C, the Western blot indicates that wild type Nog1 binds to the Lsg1-TAP particle, at least when Nog1 is over-expressed from the GAL promoter. Figure 2 also suggests that at least a subset of Lsg1-TAP particles bind Nog1, while the genetic experiments in Figure 3B clearly show that Nog1 accumulates on the Lsg1-TAP particle either when Drg1 function is impaired or in the presence of the Nog1^DN^ mutation. These data suggest that Nog1 and Lsg1 likely bind simultaneously to a subset of native cytoplasmic pre-60S particles. This point should be discussed and if appropriate, Figure 8 updated.

In comparison to WT Nog1, Nog1^DN^ accumulates on the Lsg1-TAP particle (Figure 1C, right panel). Further, WT Nog1 accumulates on the Lsg1-TAP particle in Drg1^DN^ expressing cells (Altvater et al., 2012). These data indicate that Lsg1 and Nog1 can bind simultaneously to the pre-60S in the cytoplasm. This is now discussed (subsection “Nog1 co-ordinates assembly, maturation and quality control of distant ribosomal centres”), and Figure 8 is updated accordingly.

2b) Using cryo-EM, Zhou et al., 2019, identify an early cytoplasmic pre-60S particle in which Bud20 is bound, the Nog1 CTT is in the PET, but the N-terminal and G-domains of Nog1 have apparently been displaced from the ribosome. Please discuss in the text how the SWATH-MS and genetic data presented in the current manuscript are consistent with the cryo-EM maps by Zhou et al?

We thank the reviewers for bringing this point to our notice. In the reported early pre-60S particle (Zhou et al., 2019) the terminal end of the Nog1-CTT is in the PET, and the N-terminal and G-domains are displaced from the pre-ribosome. However, Nog1 remains attached to the pre-60S via Nog1-CTT that intertwines around Rlp24. These data are consistent with the idea that Nog1 release from the pre-60S requires both Drg1-ATPase activity (through Rlp24 release) and a functional Nog1 G-domain (presumably GTPase activity). This is discussed in the text (subsection “Nog1 eviction requires a functional G-domain and Drg1-ATPase activity”).

2c) The SWATH-MS data in Figures 2 and 3 are consistent with recent cryo-EM data (Ma et al., 2017; Kargas et al., 2019) that reveal sequential recruitment of Rei1 and Reh1 into the PET. However, Reh1 is not included in the "revised model" in Figure 8. Please discuss/amend.2d) In Figure 3B, the left hand panel- consistent with the cryo-EM structures from Kargas et al., 2019, another possible interpretation of the SWATH-MS data might be that like Tif6 and Nmd3, Reh1 is simply not recycled from late pre-60S particles in the Nog1^DN^ or Drg1-deficient cells. It therefore seems unnecessary to invoke the recruitment of Reh1 and Rei1 through distinct binding sites. Please discuss. Perhaps it would be more accurate for the authors to indicate explicitly that they are proposing a revised model for the earlier steps of cytoplasmic pre-60S maturation.

The reviewers are correct that the SWATH-MS data support the model that Rei1 and Reh1 are sequentially recruited to the cytoplasmic pre-60S, and that Reh1 recycling is impaired in Nog1^DN^ expressing cells. This is now discussed in the text (subsection “The Nog1-CTT:Rlp24 complex gates PET quality control”) and included in Figure 8. We now explicitly state/discuss how our work and recent cryo-EM studies permit refinement of *early* steps that drive cytoplasmic pre-60S maturation (subsection “Nog1 co-ordinates assembly, maturation and quality control of distant ribosomal centres”).

2e) Recent work (Su et al., 2019) has identified the structure and function of Vms1 and Arb1 in RQC and mitochondrial proteome homeostasis. Please comment how the findings on Arb1 distribution in pre-60S particles in the current manuscript might relate to the identified function of Arb1 in RQC and update the text accordingly. It is inaccurate to state "structural information in the context of the ribosome is lacking".

We politely disagree with this comment: We had stated that “structural information in the context of the pre-ribosome is lacking". This statement is accurate since we do not know where Sqt1, Ecm1, Mex67, Tma16 and Arb1 are located on a pre-60S. For clarity, we have now changed “pre-ribosome” to pre-60S.

Previously, the Hinnebusch lab had shown that Arb1 contributes to the assembly of both 40S and 60S subunit (Dong et al., 2005). A recent cryo-EM structure of a translationally stalled-mature 60S subunit from the Neupert/Beckmann labs perhaps provides a clue of its location on a pre-60S. In this structure, Arb1 docks in the E-site region of 25S rRNA, which is available only when Nmd3 is released from the pre-60S. This is consistent with SWATH-MS data that Arb1 recruitment to the Lsg1-TAP:Nog1^DN^ particle is impaired. Based on all these data we speculate that Arb1 is incorporated into the pre-60S only after stable uL16 recruitment and Nmd3 release. We have incorporated these comments into the Discussion.

**Author response image 2. respfig2:** Arb1 location on a mature 60S clashes with Nmd3 binding site on a pre-60S. Cryo-EM structures of Nmd3-containing (PDB-6RZZ, Kargas et al., 2019) pre-60S and Arb1-containing (PDB-6R84, Su et al., 2019) mature 60S subunit were superimposed. Surprisingly, Arb1 binds to a region of rRNA, which overlaps with the binding site of Nmd3 on the pre-60S. We speculate that Nmd3 release allows Arb1 recruitment to the pre-60S during final cytoplasmic maturation.

2f) The Discussion should be updated to take account of recent cryo-EM data (Kargas et al., 2019) which reveals that in fact the N-terminus of Nmd3 remains fixed to Tif6 during the later steps of 60S maturation-it is the C-terminal domain that undergoes conformational rearrangement upon uL16 binding and H38 rearrangement.

We thank the reviewers for correcting our misconception(s) regarding uL16 incorporation, rearrangements in H89 rRNA and Nmd3 that drive cytoplasmic pre-60S maturation. The text has been altered accordingly (subsection “Nog1 gates stalk assembly and PTC maturation”).

2g) Figure 8A: Given that this is a "revised model for cytoplasmic maturation of the 60S ribosomal subunit", Lsg1, Reh1 and eL40 should also be included. In the figure legend, the statement "Nog1 triggers its own release together with Nsa2" does not appear to have been tested experimentally in the paper. Also, "we suggest that co-release of Nog1 and Nsa2…" Please provide evidence to support these statements or modify the text.

We have updated Figure 8A by including Lsg1, Reh1, eL40 and uL16.

Nsa2 accumulates on Lsg1-TAP in Drg1^DN^ as well as Nog1^DN^ expressing cells (Altvater et al., 2012; Figure 3C). These data are consistent with the idea that Nog1 eviction from the pre-60S precedes Nsa2 release. We have modified the text accordingly (subsection “Nog1 co-ordinates assembly, maturation and quality control of distant ribosomal centres”).

2h) Figure 8B: The typos in the legend should be corrected (H89, not H89) and the meaning of the text clarified. What rearrangement of Nmd3 is being referred to here? In the central panel, why is Rei1-CTT and not Reh1-CTT indicated? uL16 is recruited after the exchange of Rei1 for Reh1 (Kargas et al., 2019). Is this figure really necessary? It does not seem to directly relate to the work described in the manuscript and is not particularly informative.

We agree with the reviewers that Figure 8B does not provide insights into how Nog1^DN^ impairs early cytoplasmic maturation of the pre-60S. We therefore have removed Figure 8B from the manuscript, and updated Figure 8, as requested, by including Lsg1, Reh1, eL40 and uL16.